# A Small-Particle Aerosol Model of Ebolavirus Zaire Infection in Ferrets

**DOI:** 10.3390/v16121806

**Published:** 2024-11-21

**Authors:** Courtney A. Cohen, Elizabeth E. Zumbrun, James V. Writer, Luke G. Bonagofski, Charles J. Shoemaker, Xiankun Zeng, Candace D. Blancett, Christina E. Douglas, Korey L. Delp, Cheryl L. Taylor-Howell, Brian D. Carey, Suma Ravulapalli, Jo Lynne Raymond, John M. Dye, Andrew S. Herbert

**Affiliations:** 1Viral Immunology Branch, Virology Division, U.S. Army Medical Research Institute of Infectious Diseases, Frederick, MD 21702, USA; courtney.a.cohen.civ@health.mil (C.A.C.); elizabeth.e.zumbrun.civ@health.mil (E.E.Z.); john.m.dye1.civ@health.mil (J.M.D.); 2Regulated Research Administration Division, U.S. Army Medical Research Institute of Infectious Diseases, Frederick, MD 21702, USA; writerjv@yahoo.com; 3Diagnostics System Division, U.S. Army Medical Research Institute of Infectious Diseases, Frederick, MD 21702, USA; charles.j.shoemaker5.civ@health.mil (C.J.S.); candace.d.blancett.ctr@health.mil (C.D.B.); korey.l.delp.ctr@health.mil (K.L.D.); cheryl.l.taylor-howell.ctr@health.mil (C.L.T.-H.); brian.d.carey5.ctr@health.mil (B.D.C.); suma.ravulapalli.ctr@health.mil (S.R.); 4Pathology Division, U.S. Army Medical Research Institute of Infectious Diseases, Frederick, MD 21702, USA; xiankun.zeng.civ@health.mil (X.Z.); jolynne.w.raymond.civ@health.mil (J.L.R.)

**Keywords:** Ebola virus, EBOV, EVD, ferret, aerosol, intramuscular, small particle

## Abstract

The Ebola virus (EBOV) causes severe disease in humans, and animal models are needed to evaluate the efficacy of vaccines and therapeutics. While non-human primate (NHP) and rodent EBOV infection models have been well characterized, there is a growing need for an intermediate model. Here, we provide the first report of a small-particle aerosol (AE) EBOV ferret model and disease progression compared with the intramuscular (IM) EBOV ferret model. EBOV infection of ferrets by either route resulted in uniform lethality in 5–6.5 days post infection (dpi) in a dose-dependent manner, with IM-infected ferrets succumbing significantly earlier than AE-infected ferrets. EBOV disease progression differed between AE and IM routes, with significant viremia and presence of virus in target organs occurring earlier in the AE model. In contrast, significant fever, clinical signs of disease, liver pathology, and systemic inflammation occurred earlier in the IM EBOV model. Hepatocellular damage and splenic pathology were noted in both models, while pronounced lung pathology and renal impairment were exclusive to the AE and IM models, respectively. These results demonstrate that small-particle AE and IM ferret EBOV models share numerous common features with NHP and human EBOV infection by these routes and will therefore be useful for the development of vaccine and therapeutic countermeasures.

## 1. Introduction

The filoviridae family consists of six antigenically distinct species within the Orthoebolavirus genus, Zaire (EBOV), Sudan (SUDV), Bundibugyo (BDBV), Tai Forest, Reston (RSTV), and Bombali [1]. Three of these (EBOV, SUDV, and BDBV) are high-consequence threat agents capable of having a devastating impact on human health. Among the filovirus species, EBOV has been responsible for the largest number of outbreaks, cases, and deaths, the most notable of which was the 2013–2016 West Africa EBOV epidemic that caused a reported 28,616 cases with 11,310 deaths (indicating a case fatality rate of 40%) [2].

Well-characterized animal models of EBOV infection are critical for the evaluation of medical countermeasures (MCMs) like vaccines and therapeutics. In the absence of additional large outbreaks, FDA approval of new products under the Animal Rule will be necessary. Given the recent difficulties with non-human primate (NHP) supply chains, the field has begun exploring ferrets as a higher-order preclinical downselection model [3]. It is important to consider that ferrets can be more logistically complicated to house in a containment environment than rodents, requiring additional PPE to handle and extensive personnel training, similar to that required for handling NHPs. However, ferrets are easier and less expensive to acquire and maintain than NHPs and are susceptible to lethal infection by wild-type EBOV, eliminating the need for the host adaptation needed for immunocompetent rodent models [3]. While ferrets are unlikely to completely replace NHPs as the gold standard for human modeling, they may provide an invaluable downselection strategy to relieve demand and reduce overall usage of NHPs for EBOV vaccine and therapeutic studies.

The domestic ferret (*Mustela putorius furo*) is an accepted model for several viruses, including rabies virus, influenza virus, Nipah virus, respiratory syncytial virus, severe acute respiratory syndrome coronavirus [4], and more recently, filoviruses [5,6,7,8,9,10,11,12]. Ferrets are susceptible to EBOV, SUDV, BDBV, and RSTV; infection results in 100% mortality [5,6,7,8,9,10,13]. Susceptibility to a related filovirus, Marburg virus (MARV), has not been demonstrated [14,15]. Ferret models of intramuscular (IM) EBOV infection have been well described elsewhere (Table 1) [7,9,16,17,18]. EBOV challenge in ferrets via either IM or intranasal (IN) route is 100% lethal and recapitulates the indicators of EBOV disease (EVD) in humans and NHPs, including sustained fever, petechial rash, systemic viremia, viral shedding, impaired liver and kidney function, coagulopathy, and elevated markers of inflammation [3,5].

The primary mode of natural EBOV transmission among humans is direct contact with symptomatic persons or contaminated body fluids. An aerosol route of EBOV infection is not supported by epidemiological studies but may occur during aerosol-generating medical procedures [19,20,21]. However, the possibility of occasional natural exposure by the aerosol route is indicated by laboratory studies. Aerosolized EBOV remains viable for at least 90 min [22], and Harbourt et al. demonstrated that EBOV RNA could be detected in aerosol samples collected from a room housing IM EBOV-infected NHPs during a period of peak viral shedding (7 to 10 days post infection) [23]. Additionally, two uninfected NHPs in a biocontainment laboratory became infected when co-housed with aerosol (AE)-exposed NHPs, likely through either inhalational, oral, or conjunctival exposure [24].

Due to the presumed susceptibility of humans to infection from EBOV aerosols, the biodefense community is interested in the evaluation of MCM efficacy in the context of an AE exposure [24,25,26,27,28]. Limited MCMs are available for the prevention and treatment of EVD. There is currently one FDA-approved vaccine and two FDA-approved mAb treatments for parenteral exposure, but efficacy against AE challenge has not been demonstrated. A recent report has demonstrated anti-EBOV efficacy using a small molecule therapeutic (Remdesivir) via an NHP model of AE infection, indicating interest within the field in demonstrating MCM efficacy in aerosol models [29]. Murine [30,31,32,33] and guinea pig [34,35] filovirus models of aerosol challenge have been developed but require either immunodeficient animals or species-adapted filovirus strains. While AE models of filovirus infection in NHPs have been well defined [24,36,37,38,39,40,41,42], the susceptibility of ferrets to small-particle EBOV aerosols has not been established, and the course of EVD following AE exposure in ferrets has not been characterized.

In this study, we compiled data from untreated, EBOV-exposed ferrets from five independent MCM efficacy studies in order to compare and contrast small-particle AE (*n* = 20) and IM (*n* = 26) EBOV pathogenesis. To our knowledge, this is the first report of an AE EBOV infection model in ferrets, and this model can be leveraged by the field for the evaluation of MCMs.

## 2. Methods

Study Design: Data presented here represent a compilation of untreated, EBOV-infected ferrets from 5 independent MCM studies: 14 IM EBOV-exposed ferrets and 8 AE-exposed ferrets from four terminal efficacy studies, and 12 ferrets per route from a serial endpoint study in which 4 ferrets each were euthanized on days 1, 3, and 5 post exposure (46 total EBOV-exposed ferrets). These five studies included an initial lethality validation study to bring the ferret EBOV exposure model in-house and four EBOV MCM efficacy studies. The detailed breakdown of individual study design and sample analysis can be found in Table 2. The details of MCM efficacy studies are not within the scope of this manuscript and thus are not described. During the first study (*n* = 4/group), blood samples were collected 4 and 6 days post infection (dpi) for evaluation of serum viremia, serum chemistries, and hematology. However, the blood collection schedule was refined for subsequent studies (1, 3, and 5 dpi) to prevent sample loss during the acute phase, since some animals succumbed prior to 6 dpi.

Animal Welfare Statement: Ferret challenge studies were performed under an Institutional Animal Care and Use Committee (IACUC)-approved protocol in compliance with the Animal Welfare Act, Public Health Service Policy, and other applicable federal statutes and regulations relating to animals and experiments involving animals. The facility where these studies were conducted (USAMRIID) is accredited by the Association for Assessment and Accreditation of Laboratory Animal Care International and adheres to principles stated in the Guide for the Care and Use of Laboratory Animals, National Research Council, 2011.

Animals: Female ferrets (Marshall BioRecourses, North Rose, NY, USA) were between 3 and 5 months old, unspayed, and ranging from 619 to 1060 g at the time of viral exposure. Experimentally naïve ferrets were micro-chipped for temperature assessment by the vendor and pair-housed in steel metal cages throughout the experiments, which were conducted at Biosafety Level 4 (BSL-4). Water and food were provided to all ferrets ad libitum. An additional *n* = 2, unchallenged ferrets were utilized for the development of baseline lung homogenate transcriptomics as described below.

Viral Challenge: For IM exposures, ferrets were injected in the rear caudal thigh with a target dose of 1000 plaque-forming units (pfu) EBOV Zaire ’95 in 500 µL Minimum essential medium (MEM) with 5% heat-inactive fetal bovine serum (ΔFBS). The actual dose delivered averaged 778 pfu (range 125–1065 pfu). For AE exposure, unanesthetized animals were placed individually into wire mesh cages within a whole-body chamber and exposed to small-particle EBOV aerosols of approximately 3 µm generated by a 3-jet Collison nebulizer (BGI, Inc, Waltham, MA, USA) under the control of an automated exposure control system [43] containing 10 mL 3 × 10^6^ pfu/mL EBOV in MEM with 5% ΔFBS for 10 min. Four animals were exposed at a time within the whole-body chamber. While the whole-body chamber allows for a simulation of a real-life scenario of inhalation of small-particle aerosols into the deep lung, other simultaneous routes of exposure are possible by this method and cannot be ruled out (i.e., oral, ocular, cutaneous, etc.). The aerosols were sampled continuously throughout the exposure with an all-glass impinger containing MEM with 5% FBS and 0.001% antifoam A. A viral plaque assay was performed on the sampling media, and the inhaled dose was calculated based on the following formula: Vm × t × Ce, where Vm = respiratory minute volume, t = duration of the exposure, and Ce = aerosol concentration. Vm was calculated based on the body weight of each ferret using Guyton’s formula [44]. The target dose for the AE exposure was 1000 pfu, and the calculated inhaled dose was an average of 925 pfu (range 320–1763 pfu).

Clinical Observations and Humane Endpoints: Post exposure, ferrets were monitored daily for changes in weight and temperature and clinical signs of disease. Parameters included presence of petechial rash, diarrhea, nasal and ocular discharge, labored breathing, altered gait, lethargy, and spinal curvature. Monitoring for petechial rash focused only on the face and ears, as the abdominal fur was not shaved for these studies. However, given the variability in appearance of these disease indicators, scoring for signs of clinical disease was based on environmental curiosity, activity, and spinal curvature only. Assigned scores were defined as 0 = normal, active and playful, curved spine, comes to the front of the cage; 1 = less active, subdued but normal when stimulated, reluctant to come to the front of cage, curved spine; 2 = lethargic, flat back, subdued even when stimulated (triggers administration of analgesic); and 3 = bleeding, unresponsive, weak, unable to walk (endpoint that triggers humane euthanasia). Given the rapid disease progression in this model, any score above 0 triggered a second daily check at least 6 h apart (for scoring animals only).

Viral Titers: Serial log dilutions of challenge material, serum, or tissue homogenate were prepared in MEM supplemented with 5% ΔFBS, 1% gentamicin, and penicillin/streptomycin, and 200 µL was inoculated onto six-well plates containing confluent monolayers of Vero E6 cells. Following incubation at 37 °C, 5% CO_2_ for 1 h, monolayers were overlaid with 1% agarose mixed with EBME (Eagle’s basal medium) supplemented with 30 mM HEPES and 5% ΔFBS. Cells were incubated for an additional 7 days at 37 °C, and a secondary agarose overlay was applied containing 5% neutral red. Plaques were counted the following day, and titers were represented as pfu per ml. Tissue sections were homogenized in MEM media supplemented with 5% ΔFBS using M tubes and a GentleMacs homogenizer (Miltenyi Biotech, Auburn, CA United States) into 100 µg/mL suspensions, clarified at 300 *g* for 10 min, and sterile filtered with a 60 µm cell strainer. For data reporting, viremia and tissue titer figures represent a compilation of two independent animal experiments, *n* = 4 ferrets per timepoint.

Hematology and Serum Chemistry Analysis: Phlebotomy was performed on anesthetized ferrets via the vena cava, and whole blood was distributed between a 0.5 mL pediatric serum separator tube (Becton, Dickinson and Company (BD), Franklin Lakes, NJ, USA) for viremias and chemistries and a 0.5 mL pediatric K2 EDTA tube (BD) for hematology. Serum chemistry was analyzed using Piccolo General Chemistry 13 reagent discs in combination with a Piccolo Xpress point-of-care blood analyzer (Abaxis Inc., Zoetis, Union City, CA, USA). Serum was evaluated for albumin (ALB), alkaline phosphatase (ALP), alanine aminotransferase (ALT), amylase (AMY), aspartate aminotransferase (AST), blood urea nitrogen (BUN), calcium (CA), creatinine (CRE), gamma-glutamyl transferase (GGT), glucose (GLU), total bilirubin (TBIL), total protein (TP), and uric acid (UA). Whole blood was evaluated for hematology using a DxH520 hematology analyzer (Beckman Coulter Inc., Brea, CA United States). Parameters evaluated included white blood cells (WBC), lymphocytes (LYM), monocytes (MON), neutrophils (NEU), eosinophils (EO), and basophils (BA), as well as various red-blood-cell-related parameters. For data reporting, serum chemistries and hematology figures represent a compilation of four independent animal experiments (*n* = 4–12 per timepoint) for the IM group and two independent animal experiments (*n* = 4–6 per timepoint) for the AE groups.

Pathology: During necropsy, a set of tissues (lung, liver, spleen, tracheobronchial lymph nodes (TB LN)) was collected, immediately immersed in 10% neutral buffered formalin, and held within BSL-4 containment for a minimum of 21 days. After removal from containment, the tissues were trimmed, processed, embedded in paraffin, cut by microtomy, stained, and cover-slipped by routine histological methods.

Replicate sections of the peripheral and central areas of the left caudal lung lobe, spleen, liver, and TB-LN were placed on positively charged slides. After deparaffinization, rehydration, and methanol-hydrogen peroxide blocking, slides were stained using a mixture of mouse anti-EBOV GP_1,2_ (M-DA01-A5-B11, USAMRIID; 1 mg/mL) monoclonal antibody (mAb) at a dilution 1:8000 and mouse anti-EBOV VP40 mAb (B-MD04-BD07-AE11, USAMRIID; 1 mg/mL) at a dilution 1:8000, followed by a horseradish peroxidase-conjugated secondary anti-mouse polymer (K4003, Dako Agilent Pathology Solutions, Agilent Technologies, Santa Clara, CA, USA). All slides were exposed to brown chromogenic substrate, 3, 3′-diaminobenzidine (DAB) (K346711-2 and K3468, Dako Agilent Pathology Solutions, Agilent Technologies, Santa Clara, CA, USA), counterstained with hematoxylin, dehydrated, and cover-slipped. EBOV immunohistochemistry (IHC) was performed using the Dako Envision system (Dako Agilent Pathology Solutions, Agilent Technologies, Santa Clara, CA United States). The veterinary pathologist examined all slides and recorded all findings into Pristima (Xybion®, Princeton, NJ, USA). For data reporting, various microscopic pathological findings for each sample were graded as “minimal” (<10% tissue affected), “mild” (10–20% tissue affected), “moderate” (21–50% tissue affected), “marked” (51–75% tissue affected), and “severe” (>75% tissue affected). We developed a “pathology severity score” (PSS) in which these findings were assigned a weight from 1 to 5 for grades minimal to severe, respectively, and reported cumulatively for each tissue within a group. PSS data represent a compilation of two independent animal experiments; *n* = 4 ferrets per timepoint.

Transcriptomic Sample Preparation. All whole-blood and lung homogenate samples were inactivated with a 3:1 ratio of TRIzol LS (ThermoFisher Scientific Inc., Waltham, MA, USA) and then stored at −80 °C until subsequent extraction and/or analysis. RNA for NanoString analysis was manually extracted using the miRNeasy kit (QIAGEN N.V., Venlo, The Netherlands) following manufacturer’s instructions with the modification that the initial TRIzol-LS/whole blood mixture was placed in a Phasemaker phase separation tube (ThermoFisher Scientific) prior to being centrifuged at 12,000× *g* for 15 min at room temperature. The top aqueous layer was then mixed with 100% ethanol prior to transfer into a miRNeasy kit column. Total RNA was then quantified using a Qubit Fluorimeter (ThermoFisher Scientific Inc., Waltham, MA, USA).

NanoString Transcriptomic Data Collection. A customized, ferret-speciated NanoString probe panel (Bruker Inc., Billerica, MA, USA) was designed consisting of 779 gene targets, including housekeeping genes for normalization. This panel was based on the NHP host response panel originally designed by NanoString but included the addition of a number of genes associated with lung inflammation/injury, cancer, chronic obstructive pulmonary disease, and cystic fibrosis. The panel is available upon request. Hybridization buffer (70 µL) was added to 42 uL of reporter code set to make a master mixture. The master mixture (8 µL) was added to separate tubes containing 50 ng of extracted host RNA and 2 µL of the capture code set, followed by mixing. This hybridization mixture was incubated at 65 °C for 17 h followed by 4 °C until the samples were analyzed on a NanoString SPRINT^TM^ Profiler analysis system (Bruker). Data were subsequently extracted as RCC files.

Transcriptomics Data Analysis. Normalized count data were generated using criteria provided by Nanostring and extracted from RCC files using the ROSALIND Bioinformatics software suite (ROSALIND Bio, San Diego, CA, USA) [45,46,47,48,49,50,51,52,53]. ROSALIND^®^ follows the nCounter^®^ Advanced Analysis protocol of dividing counts within a lane by the geometric mean of the normalizer probes from the same lane. Housekeeping probes to be used for normalization are selected based on the geNorm algorithm as implemented in the NormqPCR R library1 (https://www.bioconductor.org/packages/release/bioc/html/NormqPCR.html, accessed on 15 July 2024). Subsequently, differential gene expression analysis was performed on these normalized counts using the R-Statistical analysis software package ver. R 4.3.0 (GNU General Public License, R Foundation) using pre-challenge whole blood as the baseline for fold-change comparisons. For lung gene expression analysis, differentially expressed genes (DEGs) were baselined to lung homogenates taken from 2 naïve ferrets. The list of DEGs for each group was subsequently analyzed by ShineyGO ver. 8.0 (South Dakota State University, Brookings, SD, USA) to generate Kyoto Encyclopedia of Genes and Genomes (KEGG) biological pathway charts and Venny ver. 2.0 (Centro Nacional de Biotecnología, Madrid, Spain) to generate VENN diagrams [54,55,56].

Statistical methodology: The survival rates at selected timepoints were compared by Fisher’s exact test, and the times to death (TTDs) were analyzed by Log-rank test for the pairwise comparison between the challenged groups. Comparisons for all weight, temperature, clinical scores, viremia, viral titers, clinical chemistries, and CBCs were implemented by a mixed model method with covariance structure and accounting for baseline. For viremia and serial lung, liver, and spleen titer data were applied a log10 transformation to better handle the distribution of data. This can improve the stability of the model. Statistical significance was determined by *p* ≤ 0.05. Analysis was implemented in SAS version 9.4. Transcriptomic statistical analysis was performed using the Benjamini-Hochberg method of estimating false discovery rates (FDR) for *p*-value adjustment using R Statistical analysis software ver. R 4.3.0 (GNU General Public License, R Foundation). Unless otherwise noted, only DEGs with a linear fold change exceeding +2 and a *p*-value ≤ 0.05 were considered statistically significant.

## 3. Results

### 3.1. Clinical Observations

Data from untreated, IM, and AE EBOV-infected ferrets from five independent experiments were analyzed in order to evaluate differences associated with the route of infection (46 total ferrets). Four of these studies (composed of 14 IM and 8 AE-exposed ferrets combined) were terminal efficacy studies, allowing for the analysis of a survival curve. The remaining two studies (12 IM- and 12 AE-exposed ferrets) were serial endpoint studies with subsets (n = 4) of animals euthanized at 1, 3, and 5 dpi. Data points from these animals were included in the temperature, weight, and clinical scoring analysis. Ferret infection with IM and AE EBOV at a target dose of 1000 pfu was 100% lethal, with a mean-time-to-death (MTD) of 5.7 and 6.1 days, respectively, representing a statistically significant difference between challenge routes (log-rank, *p* < 0.001, Figure 1A). Higher challenge doses resulted in an earlier time-to-death (death or euthanasia upon meeting scoring criteria). As such, time-to-death was significantly inversely correlated with challenge dose for the IM route alone (*p* < 0.0001, R^2^ = 0.8460) and IM and AE routes combined (*p* < 0.0001, R^2^ = 0.7640) but did not reach the significance of the AE route alone (*p* = 0.0683, R^2^ = 0.4506) by Pearson correlation.

Post exposure, ferrets were evaluated daily for weight and temperature changes, as well as clinical signs of EVD. The first disease indicator was an elevated temperature above baseline, which began at 3 dpi in IM (39.7 ± 0.2 °C vs. 39.0 ± 0.1 °C pre-exposure, *p* = 0.03) and AE (39.5 ± 0.1 °C vs. 38.8 ± 0.1 °C pre-exposure, not significant [ns]) challenged ferrets (Figure 1B). At 4 dpi, the mean peak temperature following IM and AE exposure was recorded as 41.0 ± 0.20 °C (*p* < 0.0001, compared with baseline) and 40.4 ± 0.3 °C (*p* = 0.0002, compared with baseline), respectively. Temperatures remained significantly elevated at 5 dpi (IM: 39.8 ± 0.3 °C, *p* < 0.02 compared with baseline; AE: 40.3 ± 0.3 °C, *p* = 0.0003 compared with baseline), and in the few survivors at 6 dpi, temperatures had significantly decreased from baseline (IM: 34.9 ± 1.6 °C, *p* < 0.0001 compared with baseline; AE: 37.3 ± 1.0 °C, *p* = 0.02 compared with baseline), likely indicating imminent death, and were euthanized based on humane endpoint criteria. There were no significant differences in peak temperature elevation associated with the challenge route.

Significant weight loss compared with baseline was not apparent until 5 dpi regardless of challenge route (IM: −5.3% ± 0.9%, *p* < 0.0001; AE: −5.1% ± 1.0%, *p* < 0.0001) and was even more pronounced in surviving AE ferrets at 6 dpi (IM: −4.4% ± 5.4%, *p* < 0.0004; AE: −13.5% ± 4.3%, *p* < 0.0001). There were no route-associated differences in magnitude or timing of weight loss (Figure 1C). Although additional clinical indications (rash, diarrhea, etc.) were screened for and noted sporadically, factors contributing to clinical scores > 0 were related to loss of spinal curvature and decreased levels of ferret engagement and activity, as these proved our best and most consistent metrics for disease severity. IM-exposed ferrets began exhibiting reduced activity and engagement at 3 dpi, albeit not significantly until 4 dpi (*p* = 0.03). The onset of clinical signs (clinical scores ≥ 1) in AE-exposed ferrets was delayed and reached significance at 5 dpi (*p* < 0.0001). All surviving ferrets at 6 dpi had significant clinical signs of disease and were humanely euthanized (*p* < 0.0001). The difference in onset of clinical signs between challenge routes was also statistically significant (*p* = 0.0003).

### 3.2. Serological and Hematological Analysis

To evaluate disease progression, blood samples were collected pre-infection (baseline), as well as at several timepoints (early, mid, and late) post infection to determine viral titer by plaque assay. Infectious virus was detectable in a subset of ferrets by 3 dpi in IM EBOV (50%, 6.4 × 10^3^ ± 4.2 × 10^3^ pfu/mL, *p* = 0.035) and AE EBOV (75%, 3.2 × 10^7^ ± 1.9 × 10^7^ pfu/mL, *p* < 0.001 vs. baseline). The challenge-route-associated difference at 3 dpi was also statistically significant (*p* = 0.046). By 4 dpi, it was consistently detectable in 100% of IM (1.9 × 10^5^ ± 1.2 × 10^5^ pfu/mL, *p* < 0.001) and AE (2.0 × 10^4^ ± 4.8 × 10^3^ pfu/mL, *p* < 0.0001) ferrets. Peak viremia occurred at 5 dpi in IM-exposed ferrets (4.9 × 10^7^ ± 2.3 × 10^7^ pfu/mL, *p* < 0.0001) and in AE-exposed ferrets (3.0 × 10^7^ ± 1.9 × 10^7^ pfu/mL, *p* < 0.001). By 6 dpi, only 2 IM-exposed ferrets were available for sampling, with consistently high titers compared with baseline (2.5 × 10^7^ ± 3.6 × 10^6^ pfu/mL, *p* < 0.0001) (Figure 2).

Serum chemistries were also evaluated at several timepoints post infection to monitor disease progression and evaluate challenge route-associated differences (Table 3). As expected, IM EBOV infection resulted in significantly increased levels of ALT, AST, ALP, TBIL, and GGT at 5 and 6 dpi, indicative of hepatocellular damage and consistent with previous reports (Figure 3A–E) [6,13]. AE EBOV-exposed ferrets also demonstrated significant elevation in ALT, AST, TBIL, and GGT at 5 dpi compared with pre-infection, but not ALP. There was a significant route-associated difference in ALP levels at 5 dpi between IM- and AE-exposed ferrets (*p* = 0.0009). Only IM (*p* = 0.044) induced elevated AMY, indicative of pancreatic damage (Figure 3F). 

BUN and CRE levels were significantly elevated at 5 and 6 dpi in IM EBOV-exposed ferrets compared with earlier timepoints, indicative of renal dysfunction. In contrast, these parameters were not elevated in AE-exposed ferrets, displaying a significant difference from IM-exposed ferrets at 5 dpi (*p* < 0.001) (Figure 4A,B). Other parameters measuring renal dysfunction (ALB and GLU) were significantly elevated regardless of challenge route (Figure 4D,E), and both challenge routes induced significant hypercalcemia (Figure 4F).

Whole blood was also evaluated for complete blood counts (CBCs) at various post-infection timepoints in IM and AE EBOV-exposed ferrets to monitor disease progression and challenge route-associated differences (Table 3). In IM-exposed ferrets, there was a significant reduction in overall WBCs, as well as lymphocytopenia at 4 dpi (Figure 5A,B). Significant lymphocytopenia was delayed until 5 dpi in AE-exposed ferrets, although the challenge route-associated differences were not quite statistically significant (*p* = 0.061). By 5 dpi, circulating monocytes, neutrophils, and basophils were significantly elevated over baseline levels in IM-exposed ferrets (Figure 5C–E). While AE-exposed ferrets demonstrated elevated neutrophils and basophils at 5 dpi (*p* < 0.0001 and *p* = 0.048, respectively), levels of monocytes were not significantly elevated in comparison to baseline. Elevation in both basophil and neutrophil levels was significantly different between challenge routes (*p* < 0.0001). Due to the large amount of variation in baseline values, platelet levels were only significantly reduced in AE-exposed ferrets at 5 dpi (Figure 5F).

### 3.3. Tissue Titers

At time of euthanasia, sections of lung, liver, spleen, and TB LN were harvested for analysis of infectious titer by plaque assay and pathology in IM- and AE-exposed ferrets. Tissue samples were collected either during serial endpoint studies (1, 3, and 5 dpi) or when ferrets succumbed to disease (“terminal”, 5–6.5 dpi). Ferrets that succumbed on 5 dpi were left in the “terminal” group to delineate any potential differences between those that were euthanized due to study timepoints from and those that had reached critical euthanasia endpoints.

Following AE exposure, the infectious virus disseminates more rapidly throughout tissues (Figure 6). In the lungs, viral titer was detectable at 1 dpi in 100% of AE-exposed ferrets (ranging from 6.8 × 10^3^ to 2.3 × 10^6^ pfu/mL, *p* < 0.0001) while dissemination to the spleen and liver was detectable in 2 out of 4 ferrets (ranging from 3.8 × 10^3^ to 6.5 × 10^3^, *p* = 0.006, and 1.1 × 10^4^ to 1.3 × 10^4^, *p* = 0.026, respectively) (Figure 5A–C). In contrast, no infectious virus was detectable in any of the tissues assayed following IM EBOV challenge at 1 dpi (*p* = 0.048 to < 0.0001). By 3 dpi, the infectious virus was detectable in all tissues, regardless of the inoculation route (*p* < 0001). Overall peak titers were detected in the spleen at 5 dpi in IM and AE EBOV-exposed ferrets (1.6 × 10^8^ ± 3.2 × 10^7^ pfu/mL, *p* < 0.0001 and 1.6 × 10^8^ ± 3.4 × 10^7^ pfu/mL, *p* < 0.0001, respectively). It is important to note that infectious virus was not consistently detectable in the sera until 4–5 dpi, thus, the high viral loads in lung, liver, and spleen tissues are attributed to active viral replication within tissue parenchyma, rather than circulating virus captured from the whole blood within tissues.

### 3.4. Gross Pathology

Three of the five studies compiled here included necropsy and pathological evaluation. This includes 4 ferrets per challenge route as part of a terminal study and 12 ferrets per challenge route as part of serial endpoint studies.

As described in Table 4, animals exposed to EBOV by either the IM or AE route had very few gross lesions at 1 dpi; however, two AE-exposed animals had hemorrhage and edema in the mediastinum and red discoloration of all lung lobes. At 3 dpi, nearly all IM- and AE-exposed animals had red discoloration of most lung lobes, and multiple, variably sized white foci were present in the splenic red pulp in 2/4 AE-exposed ferrets. By 5 dpi, both groups had multiple gross necropsy findings, including enlarged, discolored livers with rounded edges and friable consistency and multiple white foci within the splenic red pulp that was sometimes firmer than normal or friable. While AE-exposed ferrets had gross lesions in the lungs beginning at 1 dpi (2/4) that were persistent until animals succumbed to infection (4/4 at 3 and 5 dpi), IM-exposed ferrets did not present with gross lung lesions until 3 dpi (3/4), and these lesions were less consistent late in infection (1/4 at 5 dpi, and 0/4 in terminal ferrets). Four AE-exposed animals that did not succumb until 6 dpi had similar lesions to the four AE-exposed animals at 5 dpi with the following additions: pale renal cortices (1/4), hemorrhage in the urinary bladder (2/4), edema and hemorrhage in the mediastinum (3/4), and thymic hemorrhage (1/4).

### 3.5. Microscopic Pathology Findings

Blinded tissue slides from ferret lung, liver, spleen, and TB LNs were evaluated by a board-certified veterinary pathologist, and various microscopic pathological findings for each sample were graded as “minimal” (<10% tissue affected), “mild” (10–20% tissue affected), “moderate” (21–50% tissue affected), “marked” (51–75% tissue affected), and “severe” (>75% tissue affected). We developed a “pathology severity score” (PSS) in which these findings were assigned a weight from 1 to 5 for grades minimal to severe, respectively, and reported cumulatively for each tissue within a group.

EBOV-associated microscopic damage was detectable by 1 dpi in the liver (PSS = 2 ± 0.5) of IM-exposed ferrets and associated with minimal degeneration and necrosis of hepatocytes accompanied by low numbers of lymphocytes, macrophages, and neutrophils (Figure 7A and Figure 8A). Liver damage accumulated quickly, nearing peak PSS as early as 3 dpi in IM EBOV ferrets (PSS = 5.75 ± 1.4; Figure 7A and Figure 8B). By 5 dpi, most IM animals had at least moderate degeneration and necrosis of individual hepatocytes with accompanying inflammation composed of low to moderate numbers of macrophages, neutrophils, and intrahepatocellular, intracytoplasmic pleomorphic eosinophilic viral inclusion bodies (PSS = 6.25 ± 0.96; Figure 7A and Figure 8C). Cell swelling with vacuolation and/or loss of the cytoplasm and nuclear pyknosis is also evident in the livers of IM EBOV ferrets by 5 dpi (Figure 8C). In AE ferrets, liver damage was also detectable at 1 dpi (PSS = 2 ± 1), with minimal hepatocellular degeneration and necrosis with low numbers of inflammatory cells, similar to IM-exposed ferrets (Figure 7A and Figure 8D). By 3 dpi, there was either mild hepatocellular degeneration and necrosis or moderate to marked vacuolar degeneration of hepatocytes. In most cases, the hepatic changes were accompanied by mild neutrophilic and histiocytic inflammation (Figure 7A and Figure 8E). Peak liver damage was slower to develop in AE-exposed ferrets (5–6 dpi, PSS = 5.25–5.95; Figure 7A), but regardless of challenge route, by late-stage disease, the most common pathological findings in the liver were degeneration/necrosis and inflammation. As with IM-exposed ferrets, cell swelling with vacuolation and/or loss of the cytoplasm and nuclear pyknosis is also evident (Figure 8F).

Microscopic splenic damage was slower than liver damage to accumulate in IM EBOV ferrets (Figure 7B). At 1 dpi, extramedullary hematopoiesis (EMH), a common background finding in ferrets, was observed, but slides were considered within normal limits by the pathologist (Figure 9A). At 3 dpi, there is evidence of multifocal hyperplasia of periarteriolar macrophage sheaths (Figure 9B). By 5 dpi there was marked cell death of white pulp lymphocytes with corresponding lymphoid depletion, necrosis of the red pulp, fibrin deposition within the red pulp, infiltration of the red pulp by moderate numbers of neutrophils and/or macrophages, and extensive EMH (peak PSS = 15.3 ± 0.58; Figure 7B and Figure 9C). By contrast, In AE EBOV-exposed ferrets, splenic lymphoid depletion was apparent as early as 1 dpi (PSS = 2.5 ± 0.5; Figure 9D) and at 3 dpi (Figure 9E) in a single ferret, but there were no obvious pathological differences between the spleens of IM- and AE-exposed ferrets during late-stage EVD (AE 5 dpi, PSS = 9.5 ± 1.25; Figure 7B and Figure 9F).

Consistent with a systemic viral infection, IM EBOV did result in some lung pathology (Figure 10A,B), primarily alveolar and perivascular inflammation, due to lymphocyte, macrophage, and sometimes neutrophil infiltration that was detectable from 1 dpi (PSS = 15.5 ± 3.2; Figure 7C) and throughout the study. By 5 dpi, there is a thickening of the alveolar septa via small numbers of macrophages, neutrophils, fibrin, and edema (Figure 7C and Figure 10C). However, there are distinct challenge-route-associated patterns in the lungs. Perivascular spaces in the lung are expanded by 1 dpi in AE-exposed ferrets (Figure 7C and Figure 10D), and by 3 dpi, pathology in the lungs was greater and more varied in AE EBOV-exposed ferrets compared with IM EBOV-exposed ferrets (PSS = 35.5 ± 6.8 and 7.75 ± 4.4, respectively; Figure 7C). The most significant findings in AE EBOV ferrets at 3 dpi included increased numbers of inflammatory cells and edema in perivascular, peribronchiolar, and alveolar spaces; minimal degeneration of individual epithelial cells lining bronchioles; low numbers of neutrophils within bronchiolar lumina; and minimal-to-mild deposition of fibrin within alveolar spaces in one animal (Figure 10E). By 5 dpi in AE-exposed ferrets, there was also consistent fibrin deposition in perivascular and peribronchiolar spaces (Figure 10F). Overall, the PSS was 3.3-fold greater at 5 dpi and 4.6-fold greater at terminal collection in the lungs of AE-exposed ferrets compared with IM-exposed ferrets (Figure 7C). It is interesting to note that although there was substantial pathology found in the lungs of AE-exposed ferrets, no clinical respiratory distress was noted in these animals at any time post infection.

Similar to what was described in the lungs, IM EBOV results in minimal pathology in the TB LN (Figure 7D). Minimal-to-moderate lymphoid hyperplasia of the TB LN with low numbers of draining eosinophils in the medullary sinuses was identified at 1–3 dpi (Figure 11A,B). At 5 dpi, lymphocyte cell death and depletion of small lymphocytes within the cortex and/or paracortex, sometimes accompanied by edema, hemorrhage, and/or low numbers of macrophages within the medullary sinuses, was noted (peak PSS = 3.25 ± 0.87; Figure 11C). By contrast, AE EBOV was associated with lymphoid hyperplasia in the TB LN at 1 dpi and lymphocytolysis and lymphoid depletion with draining inflammation in the subcapsular and medullary sinuses with lymphoid hyperplasia of the follicles and/or paracortex by 3 dpi (Figure 11D,E). Peak PSS in the TB LN was demonstrated at 5 dpi, with lymphocytolysis that affected the cortex and ranged from mild to marked lymphoid depletion in the paracortical areas, macrophage and neutrophil infiltration of the subcapsular and/or medullary sinuses with fibrin deposition in some animals (Figure 11F). One animal had several intrahistiocytic intracytoplasmic viral inclusion bodies associated with the inflammation in the lymph node sinuses (PSS = 11.75 ± 1).

### 3.6. Immunohistochemical Findings

Regardless of the challenge route, there was no positive immunohistochemistry (IHC) signal in any ferret tissues at 1 dpi (Figure 12A–H). By 3 dpi, IM-exposed ferrets had detectable EBOV antigen present in the liver (4/4, Figure 12I), spleen (4/4, Figure 12J), lungs (2/4, Figure 12K), and TB LN (1/4, Figure 12L), and AE-exposed ferrets demonstrated detectable EBOV antigen in the liver (2/4, Figure 12M), spleen (3/4, Figure 12N), lungs (4/4, Figure 12O), and TB LN (4/4, Figure 12P). By day 5 (IM and AE), there was a positive signal in all examined tissues in all groups (Figure 12Q–X).

In the lungs of IM EBOV ferrets on 5 dpi, positive immunoreactivity was localized to alveolar macrophages, spindle cells, serum, and adipocytes (Figure 12S). Hepatocytes, serum, and circulating monocytes exhibited positive signals in the liver (Figure 12Q). Within the spleen, there was a positive signal within the red pulp that was mostly diffuse, so it was difficult to determine a specific cell type that was immunoreactive (Figure 12R). Within the white pulp, there was a multifocal positive signal in macrophages and reticular cells (Figure 12R). In the TB LN, there was a positive signal in macrophages and reticular cells, as well as in fibrin and serum (Figure 12T).

In the AE EBOV ferrets, similar IHC immunoreactivity was present in the examined tissues. However, positive signal was more frequent and more widely distributed in the lungs and TB LN when compared with the IM-challenged animals (Figure 12W,X). In addition to the previously discussed staining, AE ferrets also had positive signals within the smooth muscle surrounding large airways, chondrocytes, epithelial cells lining large airways, and alveolar septa (Figure 12W). Within the TB LN, there was more signal in the lymphoid follicles in reticular cells and macrophages and within the subcapsular and medullary sinuses (Figure 12X).

### 3.7. Transcriptomics

To ascertain whether the different exposure routes of EBOV in the ferret model produced different gene expression patterns over time, we used a targeted transcriptomic approach. Total RNA was extracted from whole-blood and lung homogenate samples from EBOV-infected ferrets on 1, 3, and 5 dpi (as well as two naïve ferrets to establish lung homogenate baseline expression levels), and gene expression was assessed with the NanoString platform using a targeted gene panel based on host response genes and markers of lung injury/inflammation and cancer. All gene expression fold changes were calculated relative to pre-challenge baseline samples for whole blood and naïve ferret lung tissues for lung analysis, respectively. In whole blood, we observed robust gene changes post infection relative to the pre-challenge baseline that were more abundant in the IM group (Figure 13A). At 5 dpi, IM-exposed ferrets demonstrated 197 significant DEGs, while AE-exposed ferrets only demonstrated 74 DEGs, and only 10 DEGs were shared between challenge routes (Figure 13B). Based on a KEGG pathway analysis, significant DEGs following IM exposure at 5 dpi were associated with “viral protein interaction with cytokine and cytokine receptor” and “cytokine–cytokine receptor interaction”, while DEGs associated with AE exposure were more commonly linked to the “T cell receptor signaling” and “hematopoietic cell lineage” pathways (Figure 13C). The top ten most significantly up- and downregulated genes at 5 dpi are shown in Table 5 (data from 1 and 3 dpi can be found in Appendix A).

A similar transcriptomics analysis was performed on the lung homogenates of IM- and AE-exposed ferrets. Interestingly, the highest magnitude change in gene expression in the lungs was a downregulation at 1 dpi in IM-exposed ferrets (Figure 13D). At 5 dpi, IM-exposed ferrets also demonstrated a greater number (138) of total DEGs, while AE-exposed ferrets only demonstrated 79 total DEGs, with only 24 shared between challenge routes (Figure 13E). Based on KEGG analysis, significant DEGs following IM exposure at 5 dpi were associated with the “EGFR tyrosine kinase inhibitor resistance” and “pancreatic cancer” pathways, while AE exposure was again associated with “T cell receptor signaling” (Figure 13F). The top ten most significantly up- and downregulated genes in the lung homogenate at 5 dpi are shown in Table 6 (data from 1 and 3 dpi can be found in Appendix A).

## 4. Discussion

There continues to be a need for additional anti-EBOV MCMs for the prevention and treatment of disease in humans, and given the limited number of sporadic outbreaks, advancing candidates down the developmental pipeline will likely rely heavily on animal models. While NHPs recapitulate multiple critical aspects of EVD seen in humans, studies are becoming harder to justify and accomplish. An additional downselect point between rodent studies that require species-adapted viruses or immunocompromised animals, and NHPs is warranted. While other groups have described parenteral routes of EBOV infection and disease progression in ferrets, the DoD has an interest in evaluating MCM efficacy in the context of AE exposure.

As with rodents and NHPs, ferrets present a unique set of advantages and disadvantages for MCM development. They can be more logistically complicated to house in a containment environment than rodents and require additional PPE to handle. They also require more extensive personnel training, similar to that required for handling NHPs. However, they are significantly more cost-effective than NHPs, and unlike rodents, are susceptible to wild-type viruses. There is even a small but growing number of immunological reagents available with specificity or cross-reactivity to ferrets that can be applied to gain a deeper mechanistic understanding of viral pathogenesis and MCM mechanisms of action.

Our results from IM-exposed ferrets align with previously described ferret work, in terms of the rapid disease progression, onset of temperature elevation and weight loss, time to lethality, timing of viremia, abnormalities in WBC counts, and evidence of multiorgan failure [5,7,13]. Here, we contribute to the field of ferret filovirus modeling by compiling the data from five independent ferret studies in which we evaluated MCM efficacy against AE EBOV in comparison to IM EBOV exposure. The data presented herein describe only the untreated EBOV-infected ferrets as a means of sharing our experiences and lessons learned with the broader community. We demonstrate that AE EBOV is 100% lethal in ferrets at a target dose of 1000 pfu, with elevated temperature beginning at 3 dpi, a short window of clinical indications mostly defined by the absence of activity, engagement, and spinal curvature, lethargy from 4 to 5 dpi, and an MTD of 6.1 dpi. We rarely captured additional EVD-associated clinical parameters, such as diarrhea or petechial rash, although it is possible that we missed the presence of rash, as we did not shave the animals. Further, despite “respiratory distress” being one of the parameters we evaluated in these experiments, this was not found in the AE EBOV ferrets described here. However, it should be noted that besides visual observations, no additional respiratory parameters were measured.

In addition to the specific route-associated differences described above, the timing and order of the onset of disease manifestations varied between IM and AE EBOV-infected ferrets (Figure 14). Strikingly, the virus was detected in lung, liver, and spleen tissue as early as 1 dpi in AE-exposed ferrets and not until day 3 in IM-exposed ferrets (tissues were not assessed at 2 dpi). Viremia was detected in some animals from both groups at 3 dpi, with higher viral loads detected in AE-exposed animals. Despite the higher detectable viral load in AE EBOV-exposed ferrets at 3 dpi, IM EBOV-exposed ferrets had earlier onset of significant temperature elevation, observable clinical signs, severe liver pathology, and systemic inflammation than AE-exposed animals. This suggests a more rapid disease progression following IM exposure, which resulted in the IM-exposed animals succumbing significantly earlier on average than those in the AE exposure group. Finally, while evidence of renal impairment from clinical chemistries was a more consistent hallmark of disease in IM EBOV-exposed ferrets, significant pathology in the lung and TB lymph nodes was a feature exclusive to the AE EBOV exposure group. Therefore, in addition to the significant impairment of liver function observed in both groups, there is a disparate contribution of lung and renal impairment to the overall disease burden of AE and IM EBOV-infected ferrets, respectively. While we did not observe any gross pathology in the kidneys of EBOV ferrets, and microscopic pathological analyses of this tissue were outside the scope of our current work, it is important to note that our finding of elevated BUN and CRE late in infection in IM-exposed ferrets is consistent with previous reports [7,57]. However, it would be interesting to expand on the importance of renal dysregulation in future work.

Previous work has demonstrated that parenteral EVD in ferrets follows a similar disease course to that of NHPs and humans. To our knowledge, this is the first report of AE exposure and subsequent EVD in ferrets. Here, we demonstrated a variety of clinical, serological, and pathological similarities between AE-exposed NHPs and ferrets. For instance, AE EBOV exposure in ferrets results in lymphopenia and neutrophilia post-infection. Clinical chemistries are also similar to that seen in NHPs in late-stage EVD, namely increases in liver enzymes AST, ALT, and GGT, as well as UA, and decreases in ALB, all indicative of hepatic necrosis and declining liver function [58]. Histologically, AE-EBOV is associated with alveolar histiocytosis, alveolar fibrin and multifocal fibrinoid vasculitis, congestion and lung thrombi, and alveolar edema and necrosis in both the lungs and TB LNs. In AE-exposed ferrets, EBOV antigens were present in the alveolar macrophages, spindle cells, serum, adipocytes, chondrocytes, epithelial cells lining and smooth muscle surrounding large airways, and alveolar septa, as well as intracytoplasmic viral inclusion bodies in alveolar macrophages, all reminiscent of NHP AE EBOV infection [58].

When we initiated the ferret EBOV model at USAMRIID in 2021, the goal was to utilize it for the stringent evaluation of MCMs as an additional downselection criterion prior to moving candidates forward into NHP testing. Given the effects that COVID-19 had already had on NHP cost and procurement, it was obvious that NHP studies were going to become exponentially more difficult to perform. According to the literature available, ferret model experiments indicated that a parenteral challenge with 0.01–1000 pfu EBOV was 100% lethal, with a time-to-death ranging from 5 to 7 dpi [3,5,16]. Given the (1) relatively small difference in timing of lethality despite a 5-log difference in challenge dose, (2) our desire to set a high bar for MCM efficacy, and (3) the uncertainty of an LD_50_ for AE EBOV in ferrets, we opted to conduct our studies using the 1000 pfu dose for IM and AE challenge. Although we are unable to share our MCM efficacy data at this time, it is important to note that the MCM evaluated in our studies did provide significant, complete, and consistent protection in ferrets at this dose under the appropriate conditions. While the evaluation of certain MCMs may benefit from a lower challenge dose and the hope of a slower disease progression (although the literature does indicate that this may be hard to accomplish), in our hands, it was a surmountable viral burden.

When evaluating the impact of the EBOV challenge route on host gene expression, there were marked differences in DEGs in whole blood, suggesting two distinct systemic host response profiles. As early as 1 dpi, IM EBOV resulted in the statistically significant downregulation of 15 and upregulation of 18 genes in whole blood, while AE EBOV only demonstrated significant downregulation of 2 genes and upregulation of 3 genes at this timepoint. Given the earlier MTD and more readily detectable influx of inflammatory cells associated with IM EBOV, contrasted with the readily detectable infectious virus in multiple organs at 1 dpi following AE (but not IM) EBOV exposure, it is interesting to speculate that perhaps these differing expression profiles are impacting downstream pathogenesis and/or host immune responses.

By 5 dpi, IM EBOV exposure demonstrated changes in innate signaling pathways (CXCL12, CCL4, IDO1) in the whole blood associated with antiviral, interferon I and II signaling pathways [59,60]. Based on our targeted transcriptomics analysis, IM EBOV exposure was associated with a pronounced bias towards pathways involved in cytokine signaling and inflammatory pathway activation, mirroring trends we observed in our clinical data set (i.e., increased circulating monocytes and neutrophils). It is also of interest that elevated CXCL12 transcripts have been reported in human EVD survivors, highlighting another similarity with the ferret EVD model [61]. Conversely, AE EBOV resulted in a bias towards significant DEGs associated with T cell signaling and the adaptive immune response at 5 dpi in whole blood. Transcriptomics analysis is invaluable in identifying which disease pathways are most significantly altered during the course of EVD and allows for meaningful comparisons between animal models and humans, as previously suggested by Cross et al. [62]. An understanding of the key pathways and proteins altered during EVD could help to develop mitigating therapeutic approaches. Here, we provide evidence that there are EBOV exposure route-associated transcriptomic changes that likely impact immune regulation and downstream MCM efficacy.

When considering MCM efficacy testing, it is important to note that ferrets succumb to AE EBOV exposure between 6 and 6.5 dpi, (earlier on average than in NHPs, 5–9 dpi), indicating a smaller therapeutic window post-infection. However, Bornholdt et al. demonstrated 100% protective efficacy of a mAb cocktail in IN EBOV-exposed ferrets when administering 15 mg at 3 and 6 dpi, despite demonstrating active pretreatment infection by PCR [57]. Currently, this treatment schedule represents the benchmark for evaluating anti-EBOV therapeutics in ferrets. However, for the foreseeable future, the ferret model will be best utilized as an additional downselect prior to moving candidates forward into more expensive NHP studies.

There is a bottleneck in MCM development related to NHP cost, logistical difficulties, and supply-chain limitations that is slowing the progression of preclinical products from rodent models to gold-standard NHP models. We believe that the ferret IM and AE EBOV models may provide a crucial stepping stone between rodents and NHPs and alleviate some of the strain in this field, facilitating more rapid development of MCMs.

## Figures and Tables

**Figure 1 viruses-16-01806-f001:**
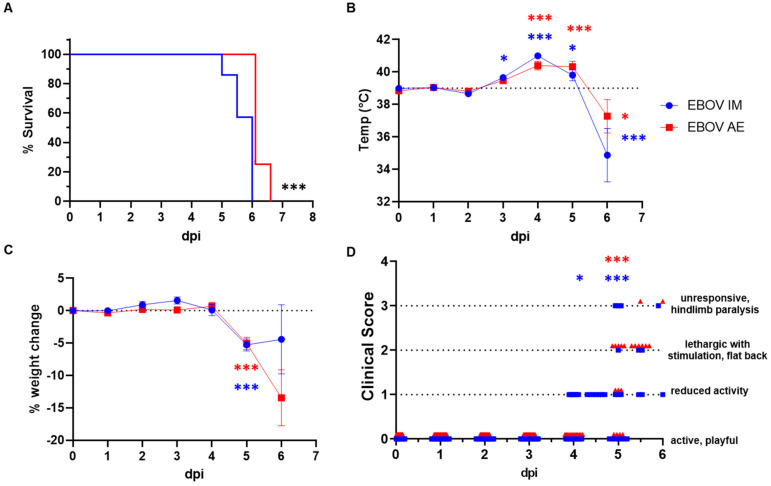
Intramuscular (IM) and aerosolized (AE) ebolavirus Zaire (EBOV) infection in ferrets. Female ferrets were exposed to a target dose of 1000 plaque forming units (pfu) EBOV, and monitored daily for survival (**A**), changes in temperature (**B**), weight (**C**) and clinical signs of disease (**D**). Dpi = days postinfection. Statistical significance is represented as * *p* < 0.05, *** *p* < 0.0005, blue represents significance between IM-treated parameter post-infection in comparison to baseline, red represents significance between AE-treated parameter post-infection in comparison to baseline, and black represents significance between challenge routes. In figure (**D**), arrows represent a consistent *p* value for subsequent timepoints.

**Figure 2 viruses-16-01806-f002:**
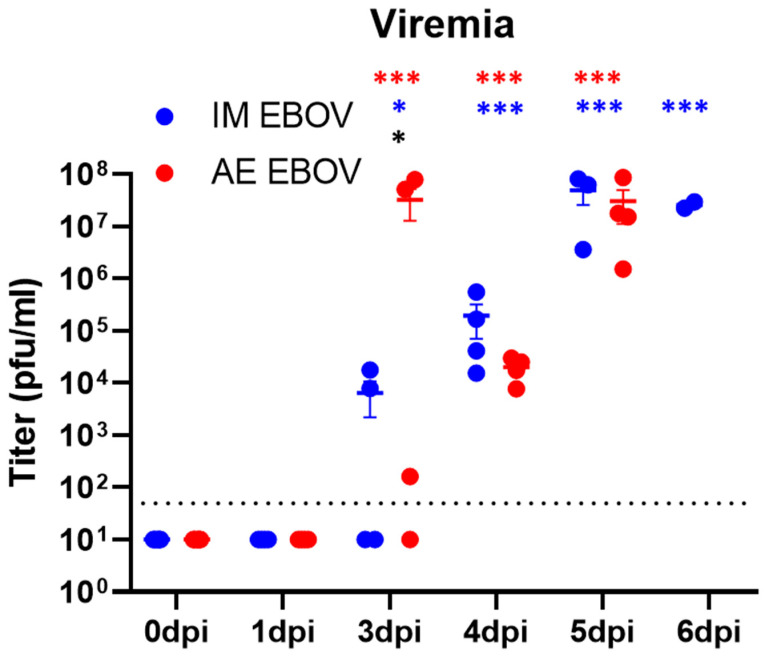
Viremia post-EBOV infection in ferrets. Female ferrets were exposed to a target dose of 1000 pfu EBOV, IM or AE. Statistical significance is represented as * *p* < 0.05, *** *p* < 0.0005, blue represents significance between IM-treated parameter post-infection in comparison to baseline, red represents significance between AE-treated parameter post-infection in comparison to baseline, and black represents significance between challenge routes.

**Figure 3 viruses-16-01806-f003:**
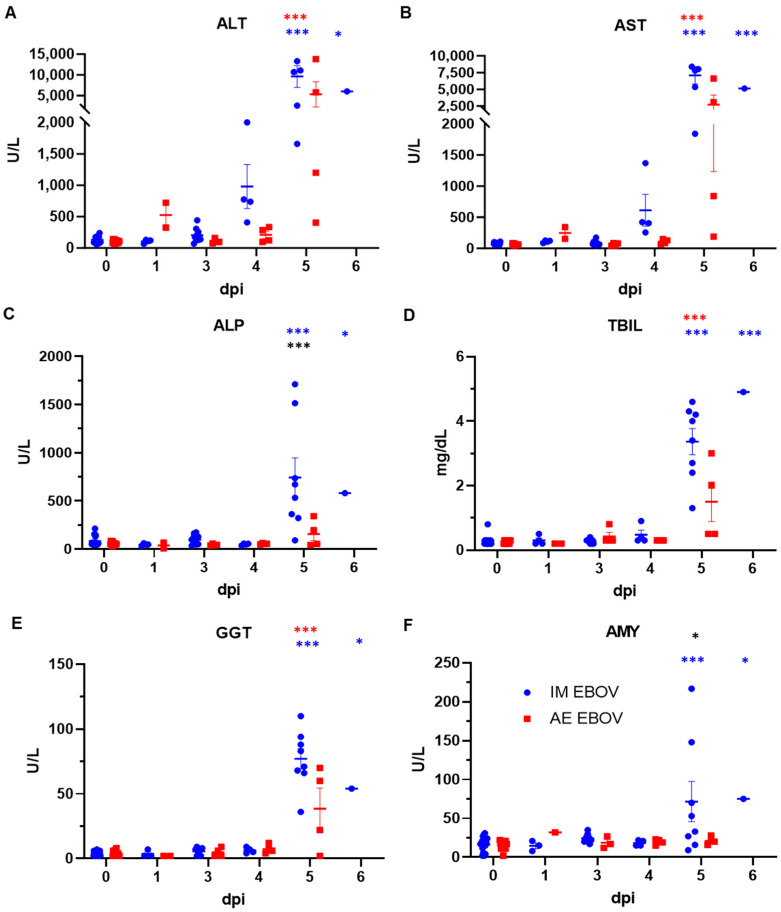
Sera chemistries following EBOV exposure in ferrets, part 1 (liver). Analysis of (**A**) Alanine aminotransferase, (**B**) Aspartate aminotransferase, (**C**) Alkaline phosphatase, (**D**) Total bilirubin, (**E**) Gamma glutamyl transferase and (**F**) Amylase. Statistical significance is represented as * *p* < 0.05, *** *p* < 0.0005, blue represents significance between IM-exposure post-infection in comparison to baseline, red represents significance between AE-exposure post-infection in comparison to baseline, and black represents significance between challenge routes.

**Figure 4 viruses-16-01806-f004:**
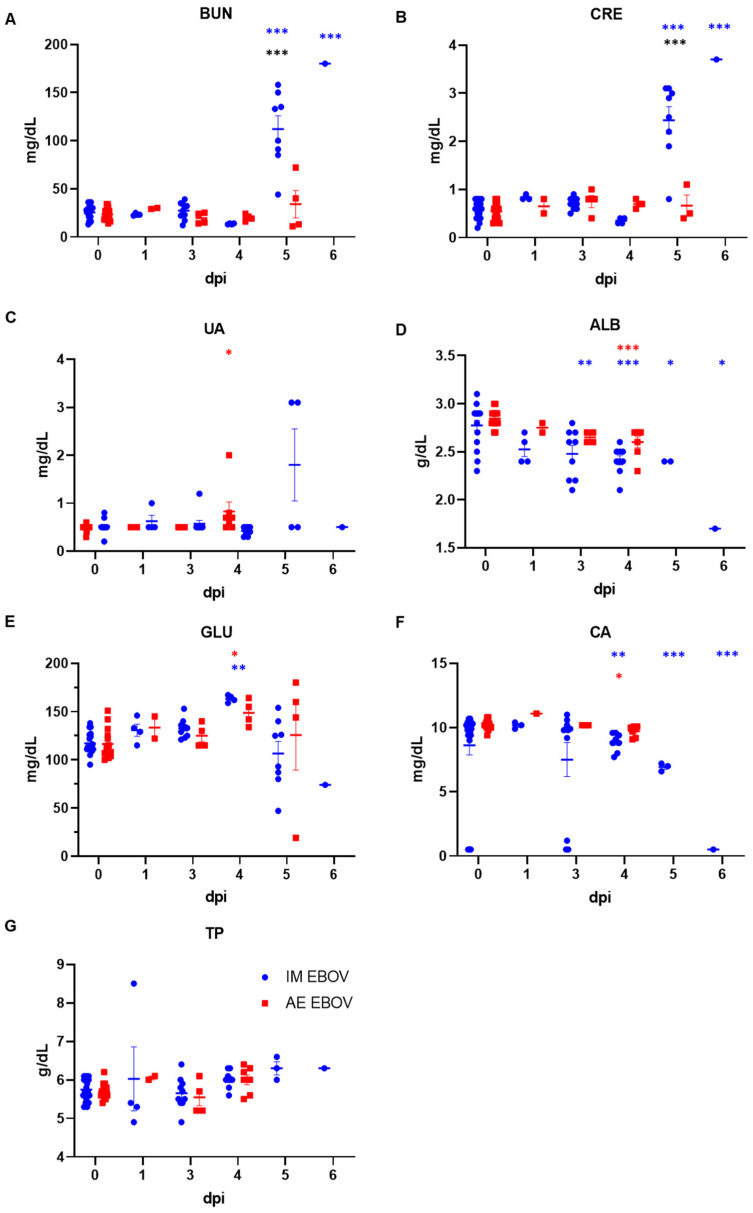
Sera chemistries following EBOV exposure in ferrets, part 2 (kidney). Analysis of (**A**) Blood urea nitrogen, (**B**) Creatine, (**C**) Uric acid, (**D**) Albumin, (**E**) Glucose, (**F**) Calcium and (**G**) Total protein. Statistical significance is represented as * *p* < 0.05, ** *p* < 0.005, *** *p* < 0.0005, blue represents significance between IM-exposure post-infection in comparison to baseline, red represents significance between AE-exposure post-infection in comparison to baseline, and black represents significance between challenge routes.

**Figure 5 viruses-16-01806-f005:**
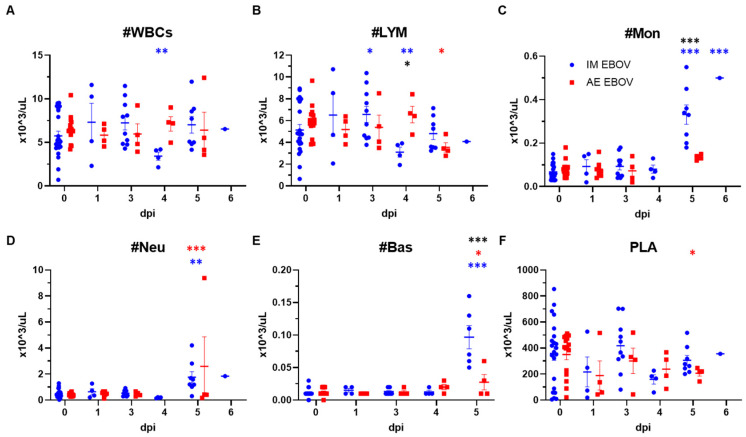
Complete blood counts in EBOV-exposed ferrets. Female ferrets were exposed to a target dose of 1000 pfu EBOV, IM or AE. Numbers of (**A**) White blood cells, (**B**) Lymphocytes, (**C**) Monocytes, (**D**) Neutrophils, (**E**) Basophils, and (**F**) Platelets per microliter whole blood are reported. Statistical significance is represented as * *p* < 0.05, ** *p* < 0.005, *** *p* < 0.0005, blue represents significance between IM-exposure post-infection in comparison to baseline, red represents significance between AE-exposure post-infection in comparison to baseline, and black represents significance between challenge routes.

**Figure 6 viruses-16-01806-f006:**
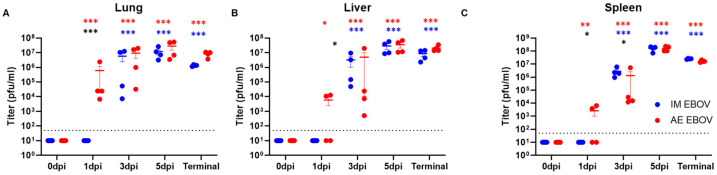
Tissue titers post-EBOV exposure in ferrets. Female ferrets were exposed to a target dose of 1000 pfu EBOV, IM or AE (n = 4 per timepoint). (**A**) Lung, (**B**) Liver, and (**C**) Spleen were homogenized and evaluated for infectious viral titer by plaque assay. Statistical significance is represented as * *p* < 0.05, ** *p* < 0.005, *** *p* < 0.0005, blue represents significance between IM-treated parameter post-infection in comparison to ferrets pre-challenge, red represents significance between AE-treated parameter post-infection in comparison to ferrets pre-challenge.

**Figure 7 viruses-16-01806-f007:**
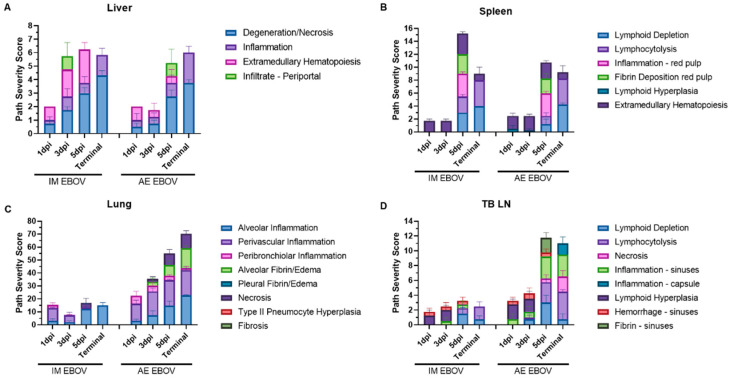
Tissue pathology severity scores in EBOV-exposed ferrets. Female ferrets were exposed to a target dose of 1000 pfu EBOV, IM or AE (n = 4 per timepoint). (**A**) Spleen, (**B**) Liver, (**C**) Lung and (**D**) Tracheobronchial lymph nodes were evaluated for EVD-associated pathology. Scores are represented as the magnitude of each finding as a sum of the whole, error represents individual ferrets.

**Figure 8 viruses-16-01806-f008:**
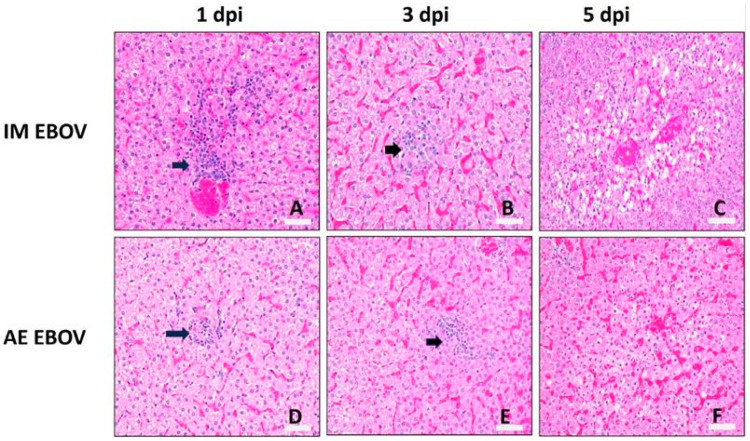
Liver histopathology. Haematoxylin and eosin (H&E) staining at 20× magnification. Photomicrographs are representative for IM EBOV (**A**–**C**) and AE EBOV (**D**–**F**) exposed ferrets 1, 3, and 5 dpi. Individual hepatocyte degeneration and infiltration by low to moderate numbers of lymphocytes and plasma cells is apparent (arrows). Scale bar, 50 µm in (**A**–**F**).

**Figure 9 viruses-16-01806-f009:**
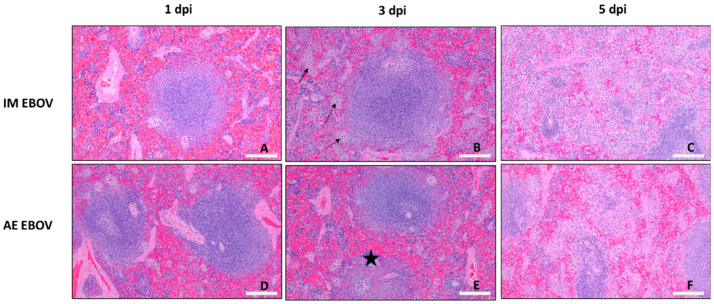
Spleen histopathology. H&E staining at 10× magnification. Photomicrographs are representative for IM EBOV (**A**–**C**) and AE EBOV (**D**–**F**) exposed ferrets 1, 3 and 5 dpi. Multifocal hyperplasia of periarteriolar macrophage sheaths is evident in IM ferrets at 3 dpi (arrows). Mild depletion of lymphocytes from a follicle evident in AE ferrets at 3 dpi (star). Scale bar, 200 µm in (**A**–**F**).

**Figure 10 viruses-16-01806-f010:**
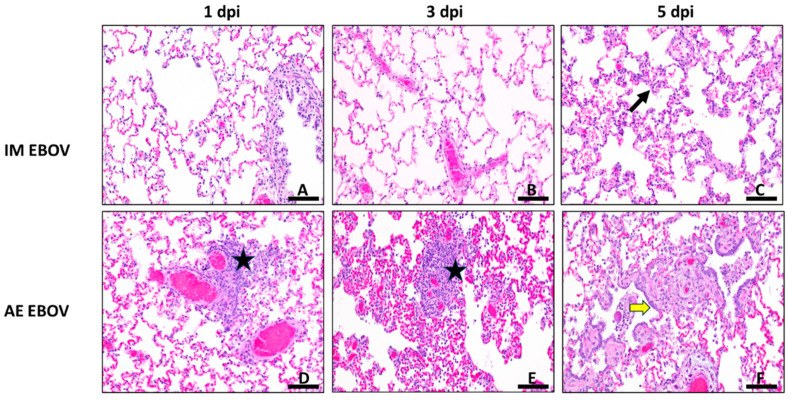
Lung histopathology. H&E staining at 20× magnification. Photomicrographs are representative for IM EBOV (**A**–**C**) an AE EBOV (**D**–**F**) exposed ferrets 1, 3, and 5 dpi. Thickening of the alveolar septa by low numbers of macrophages, neutrophils, fibrin and edema is evident at 5 dpi, IM ((**C**), arrow). Perivascular spaces are expanded by moderate numbers of macrophages, lymphocytes and neutrophils at 1 & 3 dpi, AE ((**D**,**E**), stars). Thickening of alveolar septa by collagen with type II pneumocyte hyperplasia is evident at 5 dpi, AE ((**F**), yellow arrow). Scale bar, 100 µm in (**A**–**F**).

**Figure 11 viruses-16-01806-f011:**
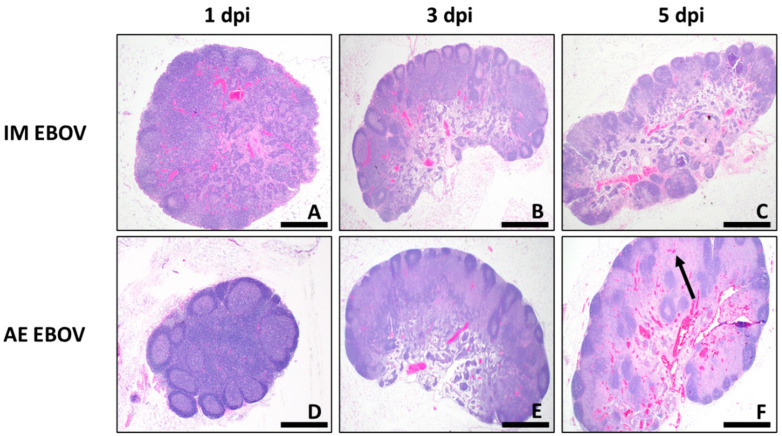
TB LN histopathology. H&E staining at 2× magnification. Photomicrographs are representative for IM EBOV (**A**–**C**) an AE EBOV (**D**–**F**) exposed ferrets 1, 3, and 5 dpi. Large areas of eosinophilia in the cortex paracortex are evident at 5 dpi ((**F**), black arrow). Scale bar, 500 µm in (**A**–**F**).

**Figure 12 viruses-16-01806-f012:**
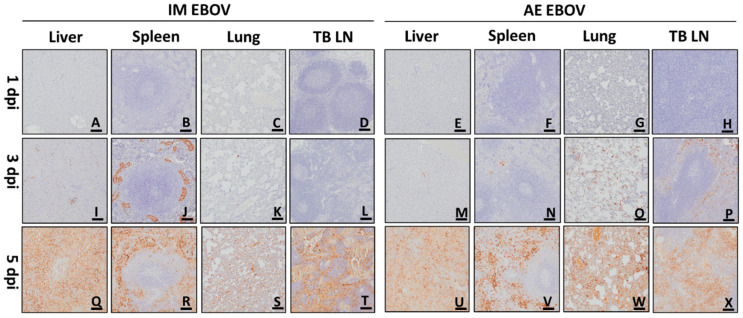
Immunohistochemistry. Representative photomicrographs of immunohistochemistry staining (brown) for IM EBOV (**A**–**D**,**I**–**L**,**Q**–**T**) and AE EBOV (**E**–**H**,**M**–**P**,**U**–**X**) exposed ferrets 1, 3, and 5 dpi. Scale bar, 100 µm in (**A**–**X**).

**Figure 13 viruses-16-01806-f013:**
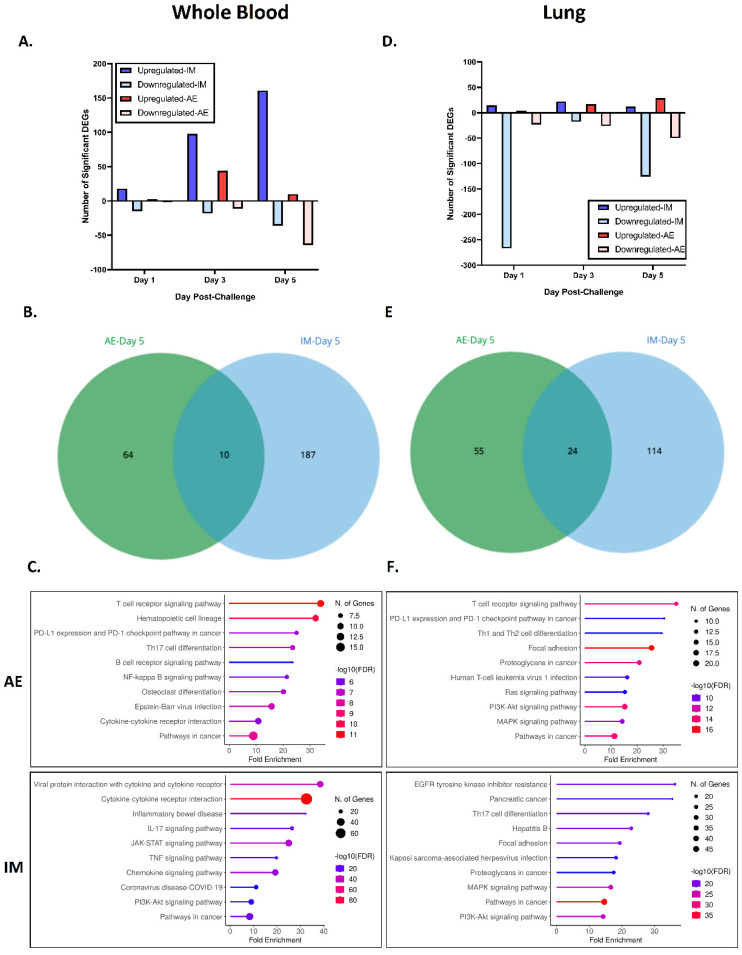
Transcriptomic analysis of whole blood and lung homogenate post-EBOV infection. Whole blood (**A**–**C**) analysis of (**A**) number of significant DEGs over time, (**B**) Challenge route-associated overlap of DEGs at 5 dpi, and (**C**) KEGG pathway analysis of significantly enriched pathways at 5 dpi. Lung homogenate analysis (**D**–**F**) of (**D**) number of significant DEGs over time, (**E**) Challenge-route associated overlap of DEGs at 5 dpi, and (**F**) KEGG pathway analysis of significantly enriched pathways at 5 dpi. All of the above differential analysis is based on aggregate analysis of 4 ferrets per group time point.

**Figure 14 viruses-16-01806-f014:**
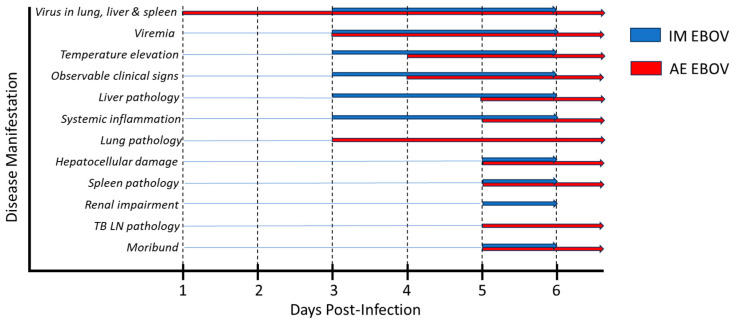
Timing and order of onset of disease manifestations in IM versus AE EBOV infected ferrets. Schematic diagram comparing the timing of onset of numerous disease states in ferrets exposed to EBOV by the IM (blue) or AE (red) routes. The earliest timepoint at which these disease traits appear is indicated, followed by a sustained presence of the trait until the animals succumbed or were humanely euthanized after meeting pre-defined endpoint criteria.

**Table 1 viruses-16-01806-t001:** Summary of published IM EBOV infection indicators in ferrets.

**Survival: 3–7 dpi**
**Clinical Signs** ➢ Fever (>39.8–40.0 °C) ➢ Increased respiration/labored breathing ➢ Lethargy/Inactivity ➢ Loose stool ➢ Unkempt appearance ➢ Gait changes ➢ Wasp-waisted ➢ Weight loss ➢ Petechial rash (some studies)
**Viremia**
**Viral shedding**
**Hepatic & renal dysfunction:** ➢ Increased alanine aminotransferase (ALT), aspartate aminotransferase (AST), & alkaline phosphatase (ALP) ➢ Increased blood urea nitrogen (BUN), creatinine (CRE) & total bilirubin (TBIL) ➢ Decreased albumin (ALB)
**Hematological changes:** ➢ Lymphopenia ➢ Granulocytosis/Neutrophilia ➢ Coagulopathy ➢ Thrombocytopenia

**Table 2 viruses-16-01806-t002:** Summary of Ferret EBOV experimental designs.

STUDY ID	# IM EBOV Ferrets	# AE EBOV Ferrets	Total Ferrets (per Exp)	Study Endpoint	Bleed Schedule (per Ferret)	Samples Evaluated for (Y/N):				
						Viremia?	Clin Chems?	Hematology?	Tissue Titers?	Path?
**Leth Exp 1**	**4**	**4**	**8**	**Terminal**	N/A	N	N	N	N	N
MCM Exp 1	4	4	8	Terminal	0, 4, 6 dpi	Y	Y	Y	Y	Y
MCM Exp 2.1/2.2 *	12	12	24	Serial (1, 3, & 5 dpi)	0 dpi & terminal	Y	Y	Y	Y	Y
MCM Exp 4 **	3	0	3	Terminal	0, 3, 5 dpi	N	Y	Y	N	N
MCM Exp 6 **	3	0	3	Terminal	0, 3, 5 dpi	N	Y	Y	N	N
**Compiled Totals:**	**26**	**20**	**46**	**Total IM ferrets represented:**		16	22	22	16	16
				**Total AE ferrets represented:**		16	16	16	16	16

Data from EBOV-exposed, untreated ferrets from five independent animal experiments has been compiled to enable comparisons between IM and AE challenge routes of infection. * MCM Exp2.1/2.2 was performed in two identical iterations spaced one month apart due to the large size (n = 80 ferrets, total) and logistical constraints within the institute. ** MCM Exps 3 & 5 related to the generation of passive sera and ferrets, included no EBOV-infected animals, and are not described in this manuscript.

**Table 3 viruses-16-01806-t003:** Timing of serum chemistry changes over baseline following IM or AE EBOV exposure in ferrets.

		IM Exposure				AE Exposure			
		*p* Value				*p* Value			
System	Analyte	3 dpi	4 dpi	5 dpi	6 dpi	3 dpi	4 dpi	5 dpi	6 dpi
Multi-System/Other	ALB (g/dL)	0.003	0.0001	0.036	<0.0001	NS	0.024	N/A	N/A
	AMY (U/L)	NS	NS	<0.0001	0.0229	NS	NS	NS	N/A
	CA (mg/dL)	NS	0.001	<0.001	0.0001	NS	NS	N/A	N/A
	GLU (mg/dL)	NS	0.0005	NS	NS	NS	0.016	NS	N/A
	TP (g/dL)	NS	NS	NS	NS	NS	NS	N/A	N/A
WBSs	# BA (×10^3^/uL)	NS	NS	<0.0001	N/A	NS	NS	0.048	N/A
	% BA (of WBCs)	NS	0.0085	<0.0001	N/A	NS	NS	<0.0001	N/A
	# LYM (×10^3^/uL)	0.0381	0.007	NS	NS	NS	NS	0.03	N/A
	% Lym (of WBCs)	NS	NS	<0.0001	0.0008	NS	NS	<0.0001	N/A
	# MON (×10^3^/uL)	NS	NS	<0.0001	N/A	NS	NS	NS	N/A
	% MON (of WBCs)	NS	0.0182	<0.0001	<0.0001	NS	NS	0.0007	N/A
	# NEU (×10^3^/uL)	NS	NS	0.0008	NS	NS	NS	0.004	N/A
	% NEU (of WBCs)	NS	NS	<0.0001	0.0106	NS	NS	0.0004	N/A
	# WBC (×10^3^/uL)	NS	0.0032	NS	NS	NS	NS	NS	N/A
Liver	ALP (U/L)	NS	NS	<0.0001	0.0166	NS	NS	NS	N/A
	ALT (U/L)	NS	NS	<0.0001	0.0125	NS	NS	<0.0001	N/A
	AST (U/L)	NS	NS	<0.0001	<0.0001	NS	NS	<0.0001	N/A
	GGT (U/L)	NS	NS	<0.0001	<0.0001	NS	NS	<0.001	N/A
	TBIL (mg/dL)	NS	NS	<0.0001	<0.0001	NS	NS	0.0001	N/A
Kidney	BUN (mg/dL)	NS	NS	<0.0001	<0.0001	NS	NS	NS	N/A
	CRE (mg/dL)	NS	NS	<0.0001	<0.0001	NS	NS	NS	N/A
	UA (mg/dL)	NS	NS	NS	NS	N/A	0.018	N/A	N/A
Coag	# PLA	NS	NS	NS	0.0248	NS	NS	0.049	N/A

*p* values represent statistically significant differences at the timepoint indicated post-EBOV exposure compared to pre-exposure baseline values. Acronym descriptions can be found in Methods (Hematology and Blood Chemistry Analysis). Orange represents a significant (*p* < 0.05) decrease from baseline, green represents a significant increase from baseline. NS = Not Significant, and N/A is used when a parameter was not captured.

**Table 4 viruses-16-01806-t004:** Frequency of gross lesions during necropsy.

Route of Exposure	Timepoint	Liver	Spleen	Lung	Other *
**IM**	1 dpi	0/4	0/4	0/4	0/4
**AE**	1 dpi	0/4	0/4	1/4	1/4
**IM**	3 dpi	0/4	0/4	3/4	0/4
**AE**	3 dpi	0/4	3/4	4/4	0/4
**IM**	5 dpi	4/4	3/4	1/4	0/4
**AE**	5 dpi	2/4	2/4	4/4	1/4
**IM**	Terminal	4/4	2/4	0/4	0/4
**AE**	Terminal	4/4	4/4	4/4	3/4

* lesions observed outside of the liver, spleen and lung.

**Table 5 viruses-16-01806-t005:** Significant differentially expressed mRNAs in ferret whole blood, 5 dpi.

	IM		AE	
	Gene	Log_2_ Diff (*p* Value)	Gene	Log_2_ Diff (*p* Value)
**Upregulated**	AOC3	7.85 (*0.045*)	NOD2	5.76 (*0.028*)
	C3	7.51 (*0.020*)	DDR2	5.356 (*0.042*)
	CCL4	7.47 (*0.020*)	CBL	5.345 (*0.0082*)
	MBL2	7.43 (*0.016*)	TREM2	5.13 (*0.035*)
	DLL1	7.28 (*0.024*)	IL1RAP	4.82 (*0.037*)
	CXCL12	7.24 (*0.021*)	ABCB1	4.67 (*0.045*)
	IDO1	7.21 (*0.025*)	TRAF4	4.49 (*0.044*)
	PLG	7.20 (*0.021*)	IL17RC	4.28 (*0.028*)
	CXCR6	7.13 (*0.024*)	VEGFA	2.82 (*0.020*)
	MC2R	7.09 (*0.022*)	TGM2	1.778 (*0.020*)
**Downregulated**	HGF	−3.83 (*0.006*)	IL32	-4.88 (*0.027*)
	NFATC2	−4.08 (*0.038*)	PDK1	−5.044 (*0.00907*)
	PLCG1	−4.13 (*0.031*)	CD3D	−5.15 (*0.0307*)
	IL17B	−4.27 (*0.026*)	FOS	−5.25 (*0.022*)
	STIM1	−4.33 (*0.034*)	CD27	−5.32 (*0.017*)
	LAMP3	−4.38 (*0.0061*)	CD74	−5.57 (*0.033*)
	CXCL14	−4.78 (*0.040*)	RASGRP1	−5.67 (*0.031*)
	POMC	−4.78 (*0.034*)	CD3G	−6.36 (*0.019*)
	FOS	−4.82 (*0.037*)	CD40LG	−6.45 (*0.047*)
	IL5RA	−5.49 (*0.021*)	CCL14	−7.73 (*0.0039*)

Top 10 of most upregulated and downregulated genes included. Inclusion criteria were a linear 2-fold difference and *p*-value ≤ 0.05.

**Table 6 viruses-16-01806-t006:** Significant differentially expressed mRNAs in ferret lung homogenate, 5 dpi.

	IM		AE	
	Gene	Log_2_ Diff (*p* Value)	Gene	Log_2_ Diff (*p* Value)
**Upregulated**	IL17B	4.88 (*0.0031*)	IL1R2	5.95 (*0.031*)
	GAST	4.69 (*0.0082*)	LEF1	5.52 (*0.0078*)
	IFNL1	4.32 (0.0068)	HAVCR2	5.52 (*0.0037*)
	TCF3	4.08 (*0.00074*)	CD28	5.48 (*0.00049*)
	IGF2	3.98 (*0.0024*)	CXCL11	5.35 (*0.0057*)
	PTX3	3.33 (*0.0043*)	CD8A	5.22 (*0.0035*)
	PTK2B	3.07 (*0.00036*)	CXCL10	4.54 (*0.020*)
	CD19	3.06 (*0.011*)	CD247	4.45 (*0.011*)
	UNG	2.84 (*0.00897*)	CD3G	4.12 (*0.041*)
	CEBPB	2.35 (*0.0097*)	CD3E	3.84 (*4.23 × 10*^−^*^5^*)
**Downregulated**	TGFBR1	−7.36 (*0.00064*)	IFNLR1	−3.95 (*0.013*)
	MFGE8	−7.57 (*0.0013*)	ESR1	−3.97 (*0.039*)
	SMAD7	−7.66 (*1.68 × 10^−^^5^*)	NT5E	−4.012 (*0.024*)
	AHR	−7.68 (*0.00032*)	CCL14	−4.19 (*0.0011*)
	NR3C1	−7.79 (*0.00037*)	PTGS2	−4.43 (*0.0012*)
	NT5E	−7.92 (*0.00015*)	GNLY	−4.45 (*0.038*)
	PLD1	−8.05 (*0.00069*)	IL17RD	−4.49 (*0.020*)
	MMP2	−8.18 (*0.00067*)	CAV1	−4.59 (*0.00036*)
	ITPR1	−8.34 (*0.00018*)	ETS1	−5.13 (*0.012*)
	PDGFRA	−8.51 (*0.0072*)	BDNF	−5.69 (*0.008*)

Top 10 of most upregulated and downregulated genes included. Inclusion criteria were a linear 2-fold difference and *p*-value ≤ 0.05.

## Data Availability

Data available upon request. Due to increased security concerns associated with federally controlled BSAT (biological select agent and toxins), data cannot be posted to a public forum. A data transfer agreement may be initiated upon request to Dr. Andrew Herbert, corresponding author, and re-use of this data will be subject to details specified in the agreement.

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
