# Peer review of "A Small-Particle Aerosol Model of Ebolavirus Zaire Infection in Ferrets"

_viruses, 2024, doi:10.3390/v16121806_

Round 1
Reviewer 1 Report
Comments and Suggestions for Authors
This manuscript covered an interesting area of work, assessing aerosol exposure of Ebola virus delivered to ferrets.
MAJOR COMMENTS
Introduction: Covering the disadvantages of the ferret model is required to provide clarity and a fair overview.
Introduction, last-but-one paragraph: only refers to comparisons with aerosolised work done in NHPs. Would include aerosol delivery in other species (e.g. A129 mice).
Methods: Would strongly suggest the authors refer to the ARRIVE guidelines for reporting animal model studies. Details on food, water, housing, clinical endpoints need adding. Also, justification for only using female animals.
Methods: The splitting of average dose and range for the survival and endpoint studies should be combined to make it easier. There groups are not separated later on.
Methods: The use of a whole body chamber needs to be discussed, as virus will not solely be inhaled as some will be on fur so ingested, etc. It was disappointing that a nose or even head-only exposure was used to give more confidence virus was indeed delivered via the respiratory route.
Methods: For the statistics, I am not sure the mixed model method with covariance structure and accounting for baseline is appropriate. More details are needed on specifics. With the small number of samples in some groups (n=4 being regular used), a non-parametric test is warranted. This has an effect on later results section, with significance values not being assigned uniformly, e.g. Figure 2 showing non-significance at 4dpi yet significance on 3dpi with more variable and lower data points. The stats around the neutrophils in the text (P<0.0001) seems influenced by a single outlier looking at Figure 5D. For Figure 6, am not sure how some results are significant (5dpi in liver and spleen), but others aren't? In legend says compared to baseline, but assume it should be uninfected ferrets? Surely all baseline/uninfected samples were negative for EBOV titres so any positive result would be significant?
Results: In a lot of the graph legends, "n=4 per timepoint" and "compilation of data from 3 independent experiments" is described. I found this difficult to understand, so would consider making clearer in the methods and then simplifying legends.
Results, Table 2: Why are there so many N/A in the parameters measured at 6 dpi? Some were measured, so sample was run. Are these below limits of detection, has the sample deteriorated..?
Results: For the graphs showing the hematology and chemistry parameters, could "normal levels", or even those of naive ferrets, be included to visualise significance of changes? The number of samples at each timepoint also varies (e.g. 2-8 in Figure 3), so why is there such a range?
Results, microscopic findings: This section was difficult to follow so needs a logical flow. The references for the figures in the text need changing. Keep Figure 7 for the quantitative values and Figure 8 for quantitative. Histology figures should have a scale bar, not magnification in the legend.
Results, IHC findings: This is much more informative than the H&E staining, but no images are given. I would highly recommend significantly reducing the earlier section and expanding this section. Perhaps combine figures (H&E and IHC panel) and including just key timepoints/findings?
Discussion: This seemed very brief, and more of a summary of the results. More comparison of the results with findings from others is essential. Figure 12 makes a good summary, but would split the top subjective into the different tissues. The abstract and discussion, the authors suggest a disparate contribution of the lung and renal involvement. I can be convinced with the former, but for the latter I am not confident that the results support this claim for the renal involvement, especially the clinical chemistry analysis.
MINOR COMMENTS
Throughout: would put hyphen in "non-human primate", "post-infection", etc.
Abstract, line 5: change to "...disease progression compared to the..."
Introduction: use updated ICTV nomenclature for virus names.
Table 1: heading refers to [5-11] but in text the references are [5-11, 14-19].
Methods: for the tissues, a 10% w/vol suspension was made. Usually reported as g/ml, so could this be adjusted?
Methods: the monoclonal antibodies used for immunohistological staining are sourced from USAMRIID where some of the authors are based. Are these available to others to allow reproducibility by others? If so, please can they provide details for accessing?
Methods: In the NanoString section, SI Table 5 is mentioned but not included in the supplementary information.
Methods: In the transcriptomics analysis section, are there references for the different approaches and software packages used?
Results: Instead of In-life, would "Clinical observations" be more descriptive? In this section, details of the studies and number of animals would fit better in the methods. In addition, correlation coefficients are provided but accompanying graphs are not, limiting interpretation.
Results, Fig 1: The title needs "Clinical signs from..." adding at the beginning. For 1D, I am not sure what the line below the 3 asterisks represents?
Results, transcriptomics: Table 4 and 5 would be better in supplementary information, and condensed, e.g. listing gene name and then putting diff and p-value in brackets instead of separate columns? For Fig 12A and 12D, would recommend having upregulated genes above the x-axis and downregulated below to make visualisation and interpretation easier to the reader.
Comments on the Quality of English Language
I would urge the authors to read through their manuscript again, as there were many small errors which made the work difficult to read.
Author Response
This manuscript covered an interesting area of work, assessing aerosol exposure of Ebola virus delivered to ferrets.
MAJOR COMMENTS
Introduction: Covering the disadvantages of the ferret model is required to provide clarity and a fair overview.
Thank you for your comment; this information is covered at the beginning of the discussion section, but we have included a description of disadvantages in the Intro as well.
Introduction, last-but-one paragraph: only refers to comparisons with aerosolised work done in NHPs. Would include aerosol delivery in other species (e.g. A129 mice).
Thank you for this suggestion. We have added a statement referencing available models of aerosolized filovirus infection in mice and guinea pigs.
Methods: Would strongly suggest the authors refer to the ARRIVE guidelines for reporting animal model studies. Details on food, water, housing, clinical endpoints need adding. Also, justification for only using female animals.
A statement has been added to the Methods addressing food and water. The statement on housing and humane endpoints are described in the Methods and have been updated for clarity. For the use of female animals: at the time these studies were run at USAMRIID, veterinary staff were recommending that only female ferrets could be pair-housed due to the increased aggressiveness of males. Therefore, due to planned study sizes and logistical constraints within the institute containment laboratory spaces, studies were designed to utilize females only.
Methods: The splitting of average dose and range for the survival and endpoint studies should be combined to make it easier. There groups are not separated later on.
Thank you for pointing this out, we have incorporated your suggested change.
Methods: The use of a whole body chamber needs to be discussed, as virus will not solely be inhaled as some will be on fur so ingested, etc. It was disappointing that a nose or even head-only exposure was used to give more confidence virus was indeed delivered via the respiratory route.
We appreciate the review pointing out this difference. We have added a statement describing that other routes of exposure (beyond that of the deep lung) cannot be ruled out. The whole-body chamber, while not specifically isolating the lungs as the only route of exposure, would represent a more realistic portrayal of a “real life” aerosol exposure.
Methods: For the statistics, I am not sure the mixed model method with covariance structure and accounting for baseline is appropriate. More details are needed on specifics. With the small number of samples in some groups (n=4 being regular used), a non-parametric test is warranted. This has an effect on later results section, with significance values not being assigned uniformly, e.g. Figure 2 showing non-significance at 4dpi yet significance on 3dpi with more variable and lower data points. The stats around the neutrophils in the text (P<0.0001) seems influenced by a single outlier looking at Figure 5D. For Figure 6, am not sure how some results are significant (5dpi in liver and spleen), but others aren't? In legend says compared to baseline, but assume it should be uninfected ferrets? Surely all baseline/uninfected samples were negative for EBOV titres so any positive result would be significant?
Using a mixed model with a covariance structure instead of a nonparametric test is often more appropriate for longitudinal or repeated measures data. Observations within the same subject (over days) are likely to be correlated rather than independent. Covariance structure allows to model the correlation between repeated measures within each subject. The significance depends on the difference and variance. We updated our statistical analysis by applying a log10 transformation to Viremias, Serial Lung, Liver and Spleen Titer to better handle its distribution. This can improve the stability of the model. Thus, we have updated the statistics for Figures 2 and 6 accordingly, and indicated the slight change in our methds section. Thank you for pointing this out, it has markedly improved our analysis.
Results: In a lot of the graph legends, "n=4 per timepoint" and "compilation of data from 3 independent experiments" is described. I found this difficult to understand, so would consider making clearer in the methods and then simplifying legends.
Thank you for your suggestion, we have altered our Methods and figure legends accordingly.
Results, Table 2: Why are there so many N/A in the parameters measured at 6 dpi? Some were measured, so sample was run. Are these below limits of detection, has the sample deteriorated..?
Thank you for catching this, it led us to repair reporting errors in our AE group. At 6dpi, there were very few EBOV-challenged ferrets remaining (n = 1 IM, and n = 2 AE). For the #BA ad %BA readouts of IM animals, the DXH520 had a “BA interference” error for that animal at that timepoint, precluding reporting of data or statistical analysis. Unfortunately this was not caught and re-run at the time of sample processing. In the AE EBOV group, there was only one animal experiment in which the collection of blood at 6dpi would have been possible (EBOV Vax Exp1), and unfortunately, 4/4 AE EBOV ferrets in this study were found dead the morning of 6dpi, which is what caused us to refine our collection timepoints to 1, 3, & 5dpi in subsequent studies. The n=2 ferrets that survived until 6.5dpi were in the Lethality Exp1, in which no blood samples were collected. Therefore, there were no measurements taken for at this timepoint in this group, and all statistical analyses have been repaired to reflect “N/A”, as is appropriate.
Results: For the graphs showing the hematology and chemistry parameters, could "normal levels", or even those of naive ferrets, be included to visualise significance of changes? The number of samples at each timepoint also varies (e.g. 2-8 in Figure 3), so why is there such a range?
Thank you for bringing this confusion to our attention, this led us to incorporate a new Table 2 in the methods section that we hope provides some granularity into the justification behind the range of data points per timepoint. Figures 3-4 (clin chemistries) and 5 (hematology) do reflect baseline values pre-challenge. Unfortunately, we did not incorporate any additional naïve ferrets into these studies.
Results, microscopic findings: This section was difficult to follow so needs a logical flow. The references for the figures in the text need changing. Keep Figure 7 for the quantitative values and Figure 8 for quantitative. Histology figures should have a scale bar, not magnification in the legend.
We updated the references for the figures in the text to better correlate the description of microscopic finds with the specific figures. Meanwhile, we added the scale bar for each individual histology images.
Results, IHC findings: This is much more informative than the H&E staining, but no images are given. I would highly recommend significantly reducing the earlier section and expanding this section. Perhaps combine figures (H&E and IHC panel) and including just key timepoints/findings?
We agree with reviewer that the immunohistochemistry data is more informative than H&E staining. Therefore, we added the immunohistochemistry data for the key timepoints for liver, spleen, lung, and TB lymph node as Figure 13 in the revised manuscript.
Discussion: This seemed very brief, and more of a summary of the results. More comparison of the results with findings from others is essential. Figure 12 makes a good summary, but would split the top subjective into the different tissues. The abstract and discussion, the authors suggest a disparate contribution of the lung and renal involvement. I can be convinced with the former, but for the latter I am not confident that the results support this claim for the renal involvement, especially the clinical chemistry analysis.
Thank you for your suggestion, we have made some additions to our discussion. In regards to the disparate renal and lung involvement in IM and AE EBOV models respectively, we do agree that it would have been beneficial to have some additional kidney function readouts to strengthen the data set, but unfortunately no significant gross pathology was noted and microscopic kidney analyses were outside the scope of this work. However, the clinical chemistry analysis demonstrates significant elevation in BUN and CRE in IM, but not AE-exposed ferrets at 5dpi compared to baseline (pre-infection) levels, the statistical analysis indicates significant route-associated differences in BUN and CRE readouts (p < 0.001) and IM-exposed ferrets also demonstrate a significantly more rapid time-to-death than AE ferrets. We believe this suggests that it’s unlikely an elevation of these parameters in AE ferrets was missed due to sample timing. Further, renal involvement in ferrets following IM EBOV exposure is consistent with previous reports.
MINOR COMMENTS
Throughout: would put hyphen in "non-human primate", "post-infection", etc.
This has been addressed, thank you.
Abstract, line 5: change to "...disease progression compared to the..."
This has been addressed, thank you.
Introduction: use updated ICTV nomenclature for virus names.
This has been corrected.
Table 1: heading refers to [5-11] but in text the references are [5-11, 14-19].
We deleted references from Table 1 heading because it is redundant to text. We corrected the references in the text to include only those referring to IM EBOV ferret infection.
Methods: for the tissues, a 10% w/vol suspension was made. Usually reported as g/ml, so could this be adjusted?
This has been adjusted as suggested by the reviewer.
Methods: the monoclonal antibodies used for immunohistological staining are sourced from USAMRIID where some of the authors are based. Are these available to others to allow reproducibility by others? If so, please can they provide details for accessing?
Unfortunately, these monoclonal antibodies are not commercially available because they were generated by investigators at USAMRIID. However, these materials are available to filovirus research field upon request and after establishment of a material transfer agreement.
Methods: In the NanoString section, SI Table 5 is mentioned but not included in the supplementary information.
This has been addressed, thank you. Since the codeset is >700 lines long, we’ve adjusted this information to be available upon request.
Methods: In the transcriptomics analysis section, are there references for the different approaches and software packages used?
Thank you for pointing this out, references have been incorporated in the relevant Methods sections.
Results: Instead of In-life, would "Clinical observations" be more descriptive? In this section, details of the studies and number of animals would fit better in the methods. In addition, correlation coefficients are provided but accompanying graphs are not, limiting interpretation.
We appreciate these recommendations. We have changed “In-life to “Clinical Observations”. The description of the studies and number of animals is also described in the methods section but feel that it needs to be stated at the front of the results section for ease of conveying the findings to the reader. Regarding the correlation coefficients, we have added “data not shown” since we are not including any accompanying graphs.
Results, Fig 1: The title needs "Clinical signs from..." adding at the beginning. For 1D, I am not sure what the line below the 3 asterisks represents?
Thank you for your comment, we have adjusted Figure 1D to attempt to clarify this. Because significance for 5dpi, 5.5dpi and 6dpi is all p<0.0005, it was difficult to incorporate all asterisks without cluttering the figure even more. We’re hoping the use of arrows and additional statement in the legend will simplify this.
Results, transcriptomics: Table 4 and 5 would be better in supplementary information, and condensed, e.g. listing gene name and then putting diff and p-value in brackets instead of separate columns? For Fig 12A and 12D, would recommend having upregulated genes above the x-axis and downregulated below to make visualisation and interpretation easier to the reader.
Thank you for the suggestions, we have incorporated these changes.
Reviewer 2 Report
Comments and Suggestions for Authors
Manuscript Number: viruses- 3198039
Title: A small particle aerosol model of Ebolavirus Zaire infection in ferrets
Summary of the study
In this paper, Cohen C.A.et al., are tackling a very important scientific concept for the Filovirus community: finding a replacement animal model to the non-human primates (NHPs) for Ebolavirus (EBOV) infection. Even if the NHPs is the gold standard model, this animal model became very difficult to work with, from the increase price of animal acquisition to the difficulty of acquiring such animals in enough numbers to properly statistically power MCM test experiments. EBOV research field needs an alternative animal model which could still reproduce hallmarks of human pathology and predict protective capacity of new MCM for future advancement in humans. In this article, the authors are comparing two infection routes, intramuscularly (IM), very commonly used in NHPs and ferrets, and aerosol (AE). So far the aerosol route was not explored in ferrets and is of great interest from exploring MCM prevention of potential EBOV AE transmission. The authors have compiled an impressive amount of data from untreated EBOV-exposed ferrets from 6 independent experiments. The authors used appropriate techniques to answer their scientific questions and data were clearly presented. A list of some major and minor comments have been provided below on which we invite the authors to provide some clarifications in order to improve the quality of the manuscript. On a side note, the manuscript I reviewed did not had the line-by-line count so unfortunately, I could not provide precise location for my comments, but I did my best to indicate as clearly as possible the section of the manuscript that I was referring to.
Major comments
1) Entire manuscript: For their studies, the authors have selected an EBOV dose of 1,000 pfu to administrate to ferrets. In the field, there is a recurrent debate about the dose administered to animal models. For example, in NHPs 1,000 pfu of EBOV is commonly used leading to 100% death between 5-12 days (depending on NHP sub-species). This EBOV dose for NHPs is already considered to be extreme and not mimicking human natural infection. Indeed, in human infections even if we do not know how many particles are necessary to mount a productive infection, it is unlikely that the transmission is occurring with a dose of 1,000 pfu. In addition, NHPs are about 3-4kg and in this study, ferrets are weighing between 0.6 to 1kg so at least 4x to 5x smaller in size but the EBOV dose used is the same. The authors are also mentioning that not all the hallmarks of NHPs and humans pathogenicity were observed in their studies.
Some clarification around the dose selection either in the introduction or in the discussion section will be helpful to the reader as currently the data are demonstrating that this current model led to a very rapid viremia spread and very rapid death with probably some pathogenicity signs missed due to the rapid evolution of the disease (system overload). If this model is to be used for testing novel MCM why selecting such stringent conditions?
2) Page 2 of 28 and Table 1: The authors are writing the following “Ferret models of parenteral EBOV infection have been well described else-where; a review of the current literature described 12 published studies encompassing data from n=73 total ferrets (Table 1) [5-11, 14-19].” and the title of Table 1 states “Summary of published parenteral EBOV infection indicators in ferrets 5-11.”
1) There is a discrepancy between the text and Table 1 title. Is Table 1 reflecting publications from #5-#11 or #5-#11 + #14-#19? Please correct accordingly.
2) The cited references are including other filovirus family members such as Sudan, Reston, Bundibugyo. Table 1, referring to references #5-#11, is stated to only include EBOV infection, so in that case, reference #9 which is exclusively on Sudan and #11 on Reston should be excluded. Please clarify.
3) Counting back only the ferrets infected with EBOV irrespective of administration routes or doses, with references #5, #6, #7, #8, #10; 77 ferrets are accounted for. We are asking the authors to clarify or correct the sentence stating that the 12 publications #5-#11 + #14-#19 are encompassing data from 73 ferrets.
4) This manuscript is focusing on IM vs AE routes and as the authors mentioned throughout the text, both routes provide different pathogenicity. While Table 1 is extremely helpful to understand what to expect upon EBOV infection in ferret, this table was made with the use of different EBOV strains, different infection routes and different doses making the interpretation of the table more complex. We are suggesting the authors to revise Table1 to account only for EBOV(Kikwit + Makona) IM route data sets.
3) Page 4 of 28 and Figures 2 and 6: In the section “Viral challenge”, the authors are describing the infection method and described that “up to four animals were exposed at a time”. For cases where more than 2 animals were exposed at a time, did those animals depicted similar time course for viral load in serum, lung, liver and spleen? Did the number of animals in the whole-body chamber impacted the number of particles that each animal could be infected with? For example, one animal received more than his neighbor? Could the authors please clarify this point in the text?
3) Page 4 of 28 and page 7 of 28: The authors are describing their materials and methods for the “Clinical Observations” and are describing parameters that were included for the observations of the ferret upon infection. “Labored breathing” is one of the parameters included, and based on data presented throughout the manuscript, the AE route is clearly impacting the lungs (Figure 6, Table 3, Figure 7.C, Figure 10). Could the authors clarify or add in the results section page 7 of 28 under “In-Life” section, if in average the AE group had more respiratory distress than the IM group?
4) Page 6 of 28: “In-Life” section, for the portion of the paragraph starting and ending with “Higher challenge doses resulted in an ….. alone (p=0.0683, R2 = 0.4506) by Pear-son correlation.” The authors are describing data that are not present in any figures or tables provided. Please add the wording “data not shown” or provide the associated figures.
5) Figures 1, 2, 3, 4, 5: A timepoint of 6dpi is represented when data are available for each of the figures but nowhere in the associated text is that timepoint described or commented on. We are asking the authors to add corresponding description/interpretation to the text for the 6 dpi timepoint for all the figures referenced.
6) Table 2 and Figures 3, 4 and 5: When comparing the statistical analysis summarized in Table 2 (AE exposure column and the 6 dpi sub-column) vs data presented in Figures 3, 4 and 5 there are several discrepancies. For example: between data reported in the figure but no statistical analysis perform (Figure 5.C #MON a red square is present at 6dpi but N/A “not captured” was reported in Table 2) or statistical analysis performed for no available data in the figure (Figure 4.A #BUN no red square present at 6dpi but NS for “Not significant” was reported in Table 2). Could the author please rectify Table 2 and associated Figures?
7) Page 6 of 28: “Immunohistochemical Findings” section, paragraph starting and finishing with “Regardless of challenge route, there ….. there was positive signal in all examined tissues in all groups.” Please indicate which Figures or Tables the authors are referring to, it will help the reader to follow along more easily.
8) Figure 12, panels A, B, C and D: We are asking the authors to clarify to the figure description if the data represent the total of 4 EBOV-infected ferrets from 1, 3 and 5 dpi experiments or if it is only a representative ferret. It is difficult to appreciate if the 4 ferrets per conditions have similar number of differentially expressed genes or if one animal is driving the observed trend.
9) Page 22 of 28: Paragraph starting and ending with “It is important to note that ferrets succumb….Currently, this treatment schedule rep-resents the benchmark for evaluating anti-EBOV therapeutics.”. I would like to mentioned to the authors that while Bornholdt Z.A. et al., (reference #14) describes the protective effect of a monoclonal antibody cocktail administrated 15mg at 3 and 6 dpi which provided complete protection in ferret, it is not the same dose that was tested and show protection toward NHPs. While ferret might be a good alternative to screen new MCM, NHPs studies might still be required to confirm therapeutics dose and effect before testing in humans.
Minor comments
1) Page 4 of 28: Please correct the reference {Hartings, 2004 #82} to be in the manuscript reference format.
2) Page 7 of 28 and Figure 1:
- Panel B: 3 dpi, it seems that the AE route should be statistically significant compared to AE baseline. Could the authors please verify the statistical analysis and correct accordingly?
- Panel C: 6 dpi, it seems that the AE route should be statistically significant compared to AE baseline. Could the authors please verify the statistical analysis and correct accordingly?
- Panel D: between 5 dpi and 6 dpi: Could the authors improve the graphical representation as there are many points failing between 5 dpi and 6 dpi and make the data very hard to interpret and visualize? from others timepoints such as 3 or 4 dpi, all animals depicted by a symbol are aligned with the dpi. The authors are also mentioning in the text “The difference in onset of clinical signs between challenge routes was also statistically significant (p = 0.0003).” Should a black star be added to the graphic as well?
3) Figures 2, 3, 4, 5: at 6 dpi there is no data set for the AE infection route. Could the authors clarify in the material and method section or in the results text section why there is no associated data set? Were all animals found dead in the cage and no blood could be retrieved from the animals to perform the different analysis?
4) Page 9 of 28: Statement “In contrast, these param-eters were not elevated in AE-exposed ferrets, displaying a significant difference from IM-exposed ferrets at 5-6 dpi (p < 0.001) (Figure 4A-B).” We are asking the authors to temper their statement as it is possible that the infection for AE is so rapid that an elevated BUN and CRE was missed due to lack of more frequent sampling before death.
5) Figure 4, panel D “ALB”: it seems that the statistical star icons are not aligned with the corresponding groups. We are asking the authors to please consider re-aligning them for clarity of the figure.
6) Figure 5, panel B “LYM”: 4 dpi, it seems that the AE route should be statistically significant compared to IM. Could the authors please verify the statistical analysis and correct accordingly?
7) Figure 6: For figures harmonization throughout the manuscript, we are suggesting the authors to move the statistical significance star icon from below the data points to above it.
Author Response
Major comments
1) Entire manuscript: For their studies, the authors have selected an EBOV dose of 1,000 pfu to administrate to ferrets. In the field, there is a recurrent debate about the dose administered to animal models. For example, in NHPs 1,000 pfu of EBOV is commonly used leading to 100% death between 5-12 days (depending on NHP sub-species). This EBOV dose for NHPs is already considered to be extreme and not mimicking human natural infection. Indeed, in human infections even if we do not know how many particles are necessary to mount a productive infection, it is unlikely that the transmission is occurring with a dose of 1,000 pfu. In addition, NHPs are about 3-4kg and in this study, ferrets are weighing between 0.6 to 1kg so at least 4x to 5x smaller in size but the EBOV dose used is the same. The authors are also mentioning that not all the hallmarks of NHPs and humans pathogenicity were observed in their studies.
Some clarification around the dose selection either in the introduction or in the discussion section will be helpful to the reader as currently the data are demonstrating that this current model led to a very rapid viremia spread and very rapid death with probably some pathogenicity signs missed due to the rapid evolution of the disease (system overload). If this model is to be used for testing novel MCM why selecting such stringent conditions?
Thank you for your comment, we have added a paragraph addressing this in our discussion section, and believe it has significantly improved the manuscript.
2) Page 2 of 28 and Table 1: The authors are writing the following “Ferret models of parenteral EBOV infection have been well described else-where; a review of the current literature described 12 published studies encompassing data from n=73 total ferrets (Table 1) [5-11, 14-19].” and the title of Table 1 states “Summary of published parenteral EBOV infection indicators in ferrets 5-11.”
1) There is a discrepancy between the text and Table 1 title. Is Table 1 reflecting publications from #5-#11 or #5-#11 + #14-#19? Please correct accordingly.
We deleted references from the Table 1 heading because it is redundant to text.
2) The cited references are including other filovirus family members such as Sudan, Reston, Bundibugyo. Table 1, referring to references #5-#11, is stated to only include EBOV infection, so in that case, reference #9 which is exclusively on Sudan and #11 on Reston should be excluded. Please clarify.
We corrected the references in the text to include only those referring to IM EBOV ferret infection.
3) Counting back only the ferrets infected with EBOV irrespective of administration routes or doses, with references #5, #6, #7, #8, #10; 77 ferrets are accounted for. We are asking the authors to clarify or correct the sentence stating that the 12 publications #5-#11 + #14-#19 are encompassing data from 73 ferrets.
While we had wished to emphasize the relatively small number of IM EBOV infected ferrets reported in the literature thus far, we decided to omit that sentence since it will likely be outdated soon after the publication of this article.
4) This manuscript is focusing on IM vs AE routes and as the authors mentioned throughout the text, both routes provide different pathogenicity. While Table 1 is extremely helpful to understand what to expect upon EBOV infection in ferret, this table was made with the use of different EBOV strains, different infection routes and different doses making the interpretation of the table more complex. We are suggesting the authors to revise Table1 to account only for EBOV(Kikwit + Makona) IM route data sets.
We corrected the references in the text of the manuscript to include only those referring to IM EBOV ferret infection, as that was the original intention of the table and the content of the table reflects this.
3) Page 4 of 28 and Figures 2 and 6: In the section “Viral challenge”, the authors are describing the infection method and described that “up to four animals were exposed at a time”. For cases where more than 2 animals were exposed at a time, did those animals depicted similar time course for viral load in serum, lung, liver and spleen? Did the number of animals in the whole-body chamber impacted the number of particles that each animal could be infected with? For example, one animal received more than his neighbor? Could the authors please clarify this point in the text?
Thank you for pointing this out. In fact, in all experiments described n=4 ferrets were placed within individual wire cages, and four wire cages at a time were placed in the whole-body chamber and exposed to aerosolized virus. I’ve adjusted this sentence in the methods section to make the meaning clearer. Unfortunately, we have no studies in which fewer animals were exposed at a time in which we can evaluate the effects of animal number on number of viral particles infected with individually. It is an interesting question that could be evaluated in future studies. There was more variability between animals in tissue titer at the early post-infection timepoints in AE-exposed animals, so it’s entirely possible that those early differences are related to differences in initial viral exposure dose.
3) Page 4 of 28 and page 7 of 28: The authors are describing their materials and methods for the “Clinical Observations” and are describing parameters that were included for the observations of the ferret upon infection. “Labored breathing” is one of the parameters included, and based on data presented throughout the manuscript, the AE route is clearly impacting the lungs (Figure 6, Table 3, Figure 7.C, Figure 10). Could the authors clarify or add in the results section page 7 of 28 under “In-Life” section, if in average the AE group had more respiratory distress than the IM group?
Thank you for your suggestion, we have added this information to the lung pathology section of Results as we felt this made more sense for a logical flow.
4) Page 6 of 28: “In-Life” section, for the portion of the paragraph starting and ending with “Higher challenge doses resulted in an ….. alone (p=0.0683, R2 = 0.4506) by Pear-son correlation.” The authors are describing data that are not present in any figures or tables provided. Please add the wording “data not shown” or provide the associated figures.
We have added “(data not shown)” since we are not including the accompanying figure with this statement.
5) Figures 1, 2, 3, 4, 5: A timepoint of 6dpi is represented when data are available for each of the figures but nowhere in the associated text is that timepoint described or commented on. We are asking the authors to add corresponding description/interpretation to the text for the 6 dpi timepoint for all the figures referenced.
Thank you for the comment, we have added commentary to the figures for 6 dpi data.
6) Table 2 and Figures 3, 4 and 5: When comparing the statistical analysis summarized in Table 2 (AE exposure column and the 6 dpi sub-column) vs data presented in Figures 3, 4 and 5 there are several discrepancies. For example: between data reported in the figure but no statistical analysis perform (Figure 5.C #MON a red square is present at 6dpi but N/A “not captured” was reported in Table 2) or statistical analysis performed for no available data in the figure (Figure 4.A #BUN no red square present at 6dpi but NS for “Not significant” was reported in Table 2). Could the author please rectify Table 2 and associated Figures?
Thank you very much for pointing this out, we have corrected the figures and table accordingly.
7) Page 6 of 28: “Immunohistochemical Findings” section, paragraph starting and finishing with “Regardless of challenge route, there ….. there was positive signal in all examined tissues in all groups.” Please indicate which Figures or Tables the authors are referring to, it will help the reader to follow along more easily.
We thank reviewer for pointing this out. To address this, we added immunohistochemistry data as Figure 12 in the revised manuscript.
8) Figure 12, panels A, B, C and D: We are asking the authors to clarify to the figure description if the data represent the total of 4 EBOV-infected ferrets from 1, 3 and 5 dpi experiments or if it is only a representative ferret. It is difficult to appreciate if the 4 ferrets per conditions have similar number of differentially expressed genes or if one animal is driving the observed trend.
Thank you for your comment, we have adjusted the figure legend to clarify this point.
9) Page 22 of 28: Paragraph starting and ending with “It is important to note that ferrets succumb….Currently, this treatment schedule rep-resents the benchmark for evaluating anti-EBOV therapeutics.”. I would like to mentioned to the authors that while Bornholdt Z.A. et al., (reference #14) describes the protective effect of a monoclonal antibody cocktail administrated 15mg at 3 and 6 dpi which provided complete protection in ferret, it is not the same dose that was tested and show protection toward NHPs. While ferret might be a good alternative to screen new MCM, NHPs studies might still be required to confirm therapeutics dose and effect before testing in humans.
Thank you for your comment, we have incorporated an additional clarifying statement to our discussion.
Minor comments
1) Page 4 of 28: Please correct the reference {Hartings, 2004 #82} to be in the manuscript reference format.
Repaired, thank you.
2) Page 7 of 28 and Figure 1:
- Panel B: 3 dpi, it seems that the AE route should be statistically significant compared to AE baseline. Could the authors please verify the statistical analysis and correct accordingly?
The associated p value here was 0.09, we believe that this difference didn’t quite reach statistical significance because there was a smaller number of ferrets within this group (as described in Table 2).
- Panel C: 6 dpi, it seems that the AE route should be statistically significant compared to AE baseline. Could the authors please verify the statistical analysis and correct accordingly?
Thank you for pointing this out – in fact both IM and AE group changes are statistically significant from baseline, and we failed to report it in the original manuscript. The figure has been updated accordingly.
- Panel D: between 5 dpi and 6 dpi: Could the authors improve the graphical representation as there are many points failing between 5 dpi and 6 dpi and make the data very hard to interpret and visualize? from others timepoints such as 3 or 4 dpi, all animals depicted by a symbol are aligned with the dpi. The authors are also mentioning in the text “The difference in onset of clinical signs between challenge routes was also statistically significant (p = 0.0003).” Should a black star be added to the graphic as well?
We’ve attempted to clarify this in Panel D without cluttering the graph too badly. Twice daily checks were triggered when animals started exhibiting clinical signs of disease base on our scoring criteria, but we’ve adjusted scale and added hash tags to second check timepoints on the X axis to illustrate this more effectively. We’ve also removed the hash bar, and incorporated arrows instead to try to depict the same p value is consistent for more than one timepoint without over-cluttering the graph.
3) Figures 2, 3, 4, 5: at 6 dpi there is no data set for the AE infection route. Could the authors clarify in the material and method section or in the results text section why there is no associated data set? Were all animals found dead in the cage and no blood could be retrieved from the animals to perform the different analysis?
Unfortunately, this is correct, in early studies we were still learning about the ferret AE EBOV model and this cohort were all found dead the morning of 6dpi. This was our justification for 1) increasing checks once animals begin scoring, and 2) altering our bleed schedule to odd days post-infection, rather than even.
4) Page 9 of 28: Statement “In contrast, these param-eters were not elevated in AE-exposed ferrets, displaying a significant difference from IM-exposed ferrets at 5-6 dpi (p < 0.001) (Figure 4A-B).” We are asking the authors to temper their statement as it is possible that the infection for AE is so rapid that an elevated BUN and CRE was missed due to lack of more frequent sampling before death.
We disagree with this statement, as we were able to report daily changes in sera chemistries for both IM and AE cohorts from pre-infection to 5dpi, and mean-time-to-death in IM animals was actually slightly (but significantly) shorter than in AE animals. While there’s always the possibility that there are acute changes occurring that we may miss, we don’t find these particular data points to be more unreliable than any other.
5) Figure 4, panel D “ALB”: it seems that the statistical star icons are not aligned with the corresponding groups. We are asking the authors to please consider re-aligning them for clarity of the figure.
We agree and have repaired, thank you for catching.
6) Figure 5, panel B “LYM”: 4 dpi, it seems that the AE route should be statistically significant compared to IM. Could the authors please verify the statistical analysis and correct accordingly?
We agree and have repaired, thank you for catching.
7) Figure 6: For figures harmonization throughout the manuscript, we are suggesting the authors to move the statistical significance star icon from below the data points to above it.
We agree and have repaired, thank you for catching.
Reviewer 3 Report
Comments and Suggestions for Authors
I recommend publication, with suggested narrative improvements to be incorporated at Editors’ and Authors’ discretion. Authors report on pathogenetic similarities and differences between intramuscular and aerosol infections with Ebolavirus Zaire (EBOV) in ferrets, both routes being uniformly lethal. The rationale for the comparison is to illuminate the pathway to licensure for medical countermeasures (MCM) against (EBOV) when effectiveness is best demonstrated against both routes independently. Here, data are consolidated from control groups of animals infected (“challenged”) by either intramuscular or aerosol route, even while the primary aims of those studies (evaluating effectiveness of various MCM strategies) might have been inconclusive for innumerable reasons including small group sizes. As such, the consolidated data reported here remain useful for others in this research field and for later reference when MCM results are more ready for regulatory consideration. If the manuscript is more than a little laborious to read in depth, it was surely even more laborious to write, and for this I congratulate the authors.
Citation of the challenge dose could use some embellishment, if minor. I understand the rationale for use of a target dose of 1000 PFU, but most readers do not, especially when informed that the LD50 may be as low as 1 PFU (is this known?) as it is in NHP studies; in many quarters (investigators more knowledgeable of other agents) a challenge dose approaching 1000 LD50 strikes strikes the reader as extraordinarily high, a nearly impossible hurdle to clear for MCM. Can this rationale be described briefly, or expediently referenced? I recall at least one reference in which a dose between 100 and 1000 PFU was rationalized for NHP, which would seem to give license for the same dose for ferrets.
In describing reagents (e.g. monoclonal antibodies), it would be helpful for future context to at least estimate the concentration of the starting material, rather than only the dilution made.
Can the authors illuminate how the differentially expressed mRNA data in ferrets can helpfully inform an approach to MCM choices for human use? I’m happy to have the data included for hypothesis-building, documentation, and for comparison with other labs’ results, but both the meaning and utility escape me. I will pardon authors if this explanatory task is too daunting.
An observation: References were helpful, and one of them drew my attention to one I had not seen, and which strikes me as interesting, extraordinary, and also highly problematic for the chimeric virus contructed therein: Schiffman, Z., et al., The Inability of Marburg Virus to Cause Disease in Ferrets Is Not Solely Linked to the Virus Glycoprotein. J Infect Dis, 2023. 228(Supplement_7): p. S594-S603. I am skeptical the risks of this approach are worth the knowledge obtainable. There is no apparent relationship to this manuscript, but I'm pleased the reference was included.
Finally, I do not intend disrespect or stigma when I rate some elements of a manuscript as “average”. Standards are and should be high. The most average of research can be highly useful, and for work undertaken in high-containment laboratories, it is uncommonly difficult logistically, and dangerous to hands-on staff.
Author Response
I recommend publication, with suggested narrative improvements to be incorporated at Editors’ and Authors’ discretion. Authors report on pathogenetic similarities and differences between intramuscular and aerosol infections with Ebolavirus Zaire (EBOV) in ferrets, both routes being uniformly lethal. The rationale for the comparison is to illuminate the pathway to licensure for medical countermeasures (MCM) against (EBOV) when effectiveness is best demonstrated against both routes independently. Here, data are consolidated from control groups of animals infected (“challenged”) by either intramuscular or aerosol route, even while the primary aims of those studies (evaluating effectiveness of various MCM strategies) might have been inconclusive for innumerable reasons including small group sizes. As such, the consolidated data reported here remain useful for others in this research field and for later reference when MCM results are more ready for regulatory consideration. If the manuscript is more than a little laborious to read in depth, it was surely even more laborious to write, and for this I congratulate the authors.
Citation of the challenge dose could use some embellishment, if minor. I understand the rationale for use of a target dose of 1000 PFU, but most readers do not, especially when informed that the LD50 may be as low as 1 PFU (is this known?) as it is in NHP studies; in many quarters (investigators more knowledgeable of other agents) a challenge dose approaching 1000 LD50 strikes strikes the reader as extraordinarily high, a nearly impossible hurdle to clear for MCM. Can this rationale be described briefly, or expediently referenced? I recall at least one reference in which a dose between 100 and 1000 PFU was rationalized for NHP, which would seem to give license for the same dose for ferrets.
Thank you for your comment, we have addressed this in the discussion section, and believe it significantly improves the manuscript.
In describing reagents (e.g. monoclonal antibodies), it would be helpful for future context to at least estimate the concentration of the starting material, rather than only the dilution made.
We added the initial concentration of monoclonal antibodies.
Can the authors illuminate how the differentially expressed mRNA data in ferrets can helpfully inform an approach to MCM choices for human use? I’m happy to have the data included for hypothesis-building, documentation, and for comparison with other labs’ results, but both the meaning and utility escape me. I will pardon authors if this explanatory task is too daunting.
Thank you for asking for this clarification. The following text has been added to convey the rationale for including mRNA data:
“Transcriptomics analysis aids in identifying which disease pathways are most significantly altered during the course of EVD in ferrets and can allow for meaningful comparisons to EVD in NHPs and humans, as also reported by Cross et al [48]. An understanding of the key pathways and proteins altered during EVD could help with developing therapeutic approaches that ameliorate such changes.”
Round 2
Reviewer 2 Report
Comments and Suggestions for Authors
Thank you so much to the authors for providing detailed and clear responses to my comments. I am satisfied with the editions provided to the manuscript.
Best,
Author Response
Comment: Please recheck the figure and table numbers match the citations in the text
Reply: After a thorough review of our revised manuscript, we noticed that Figure 2 was mistakenly deleted from the Word version of the revised manuscript, though it remained in the PDF file of Main Figures. This caused figure citations in the text to be inaccurate. We re-inserted Figure 2 in the "8 Nov 24" version of the revised manuscript and have uploaded it here. We hope this addresses the Academic Editor's concerns.